# Evolution of antibody immunity following Omicron BA.1 breakthrough infection

Chengzi I. Kaku [1,2], Tyler N. Starr [3,4], Panpan Zhou [2,5,6], Haley L. Dugan[1], Paul Khalifé[1], Ge Song [2,5,6], Elizabeth R. Champney[1], Daniel W. Mielcarz [7], James C. Geoghegan[1], Dennis R. Burton [2,5,6,8], Raiees Andrabi [2,5,6], Jesse D. Bloom [3,9,10] & Laura M. Walker[11,12] ✉

Understanding the longitudinal dynamics of antibody immunity following heterologous SAR-CoV-2 breakthrough infection will inform the development of next-generation vaccines. Here, we track SARS-CoV-2 receptor binding domain (RBD)-specific antibody responses up to six months following Omicron BA.1 breakthrough infection in six mRNA-vaccinated individuals. Cross-reactive serum neutralizing antibody and memory B cell (MBC) responses decline by two- to four-fold through the study period. Breakthrough infection elicits minimal de novo Omicron BA.1-specific B cell responses but drives affinity maturation of pre-existing cross-reactive MBCs toward BA.1, which translates into enhanced breadth of activity across other variants. Public clones dominate the neutralizing antibody response at both early and late time points following breakthough infection, and their escape mutation profiles predict newly emergent Omicron sublineages, suggesting that convergent antibody responses continue to shape SARS-CoV-2 evolution. While the study is limited by our relatively small cohort size, these results suggest that heterologous SARS-CoV-2 variant exposure drives the evolution of B cell memory, supporting the continued development of next-generation variant-based vaccines.

The emergence and global spread of the SARS-CoV-2 Omicron BA.1 variant in late 2021 resulted in the largest surge in COVID-19 caseloads to date[1]. While first-generation COVID-19 vaccines induced high levels of protection against pre-Omicron variants, the extensive immune evasiveness of Omicron resulted in significantly reduced vaccine efficacy and durability following both primary and booster immunization[2–5]. Moreover, antigenically drifted sublineages of Omicron (e.g., BA.4/5, BA.2.75.2, BA.4.6, BQ.1.1, and XBB) continue to emerge and supplant prior subvariants[4,6,7]. The high prevalence of Omicron breakthrough infections led to the development of Omicron variant-based booster mRNA vaccines, and emergency use authorization was granted based on short-term immunogenicity data[8–11]. Thus, there is an urgent need to understand if and how secondary exposure to antigenically divergent variants, such as Omicron, shape SARS-CoV-2-specific B cell memory.

We and others have previously reported that the acute B cell response following Omicron BA.1 breakthrough infection is dominated

[1]Adimab, LLC, Lebanon, NH 03766, USA. [2]Department of Immunology and Microbiology, The Scripps Research Institute, La Jolla, CA 92037, USA. [3]Basic Sciences Division, Fred Hutchinson Cancer Center, Seattle, WA 98109, USA. [4]Department of Biochemistry, University of Utah School of Medicine, Salt Lake City, UT 84112, USA. [5]IAVI Neutralizing Antibody Center, The Scripps Research Institute, La Jolla, CA 92037, USA. [6]Consortium for HIV/AIDS Vaccine Development (CHAVD), The Scripps Research Institute, La Jolla, CA 92037, USA. [7]Dartmouth Cancer Center, Geisel School of Medicine, Lebanon, NH 03766, USA. [8]Ragon Institute of Massachusetts General Hospital, Massachusetts Institute of Technology, and Harvard University, Cambridge, MA 02139, USA. [9]Department of Genome Sciences, University of Washington, Seattle, WA 98109, USA. [10]Howard Hughes Medical Institute, Seattle, WA 98109, USA. [11]Invivyd Inc., Waltham, MA 02451, USA. [12]Present address: Moderna, Inc., Cambridge, MA, USA. ✉e-mail: laura.walker@modernatx.com

by re-activated memory B cells induced by mRNA vaccination[12–15]. Consistent with these findings, booster immunization with Omicron variant-containing mRNA vaccines induces modestly higher (up to fivefold) peak serum-neutralizing antibody responses compared with booster vaccination with the original mRNA vaccines based on the Wuhan-1 strain[8,9,11,16,17]. Although these studies provide evidence for antigenic imprinting in the early B cell response following Omicron breakthrough infection, if and how this response evolves over time remains unclear.

To address these questions, we longitudinally profile SARS-CoV-2-specific serological and memory B responses in mRNA-vaccinated donors up to 6 months following BA.1 breakthrough infection. Serum-neutralizing antibodies and circulating RBD-specific memory B cells remained detectable up to 6 months post-infection. While the acute B cell response following BA.1 breakthrough infection was dominated by vaccine-induced cross-reactive clones that exhibited preferential WT binding and neutralization, antibodies isolated from the same donors 5 to 6 months post-infection accumulated additional somatic mutations and displayed enhanced BA.1 recognition at the expense of WT binding. Unmutated common ancestors of BA.1-preferring antibodies isolated at the late time point displayed preferential WT binding, providing evidence for the evolution of pre-existing WT-induced memory B cells toward BA.1. De novo BA.1-specific B cell responses only comprised a small fraction of the total RBD-directed response at both time points studied. The results are consistent with prolonged maturation of B cell memory following BA.1 breakthrough infection and suggest that heterologous variant exposure may broaden SARS-CoV-2-specific memory B cell repertoires.

## Results

### Serum-neutralizing antibody titers modestly decline over the course of 6 months following BA.1 breakthrough infection

We initially characterized the antibody response to SARS-CoV-2 in a cohort of seven mRNA-1273 vaccinated donors 14 to 27 days (median = 23 days) after BA.1 breakthrough infection[12]. To study the evolution of this response, we obtained blood samples from six of the seven participants at a follow-up appointment 4 to 6 months (median = 153 days) post-infection (Fig. 1a and Supplementary Tables 1, 2). Three of the six donors experienced infection after two-dose mRNA-1273 vaccination, while the remaining three donors were infected after a third booster dose. None of the donors reported a second breakthrough infection between the two sample collection time points.

To evaluate serum neutralization breadth and potency, we tested the plasma samples for neutralizing activity against SARS-CoV-2 D614G, emergent variants (BA.1, BA.2, BA.4/5, BA.2.75, Beta, and Delta), and the more evolutionarily divergent sarbecovirus SARS-CoV, in a murine-leukemia virus (MLV)-based pseudovirus assay. Paired comparisons within each participant revealed that serum-neutralizing titers against D614G declined by a median of 4.8-fold at 5- to 6- months post-infection relative to those observed within 1-month post-infection (Fig. 1b). Correspondingly, we observed lower serum-neutralizing titers against Omicron subvariants (2.8 to 3.9-fold, respectively), Beta (1.6-fold), Delta (3.8-fold), and SARS-CoV (3.1-fold) at the 5- to 6-month time point relative to the early time point (Fig. 1b). Despite this waning of neutralizing antibody titers over time, all of the donor sera displayed detectable neutralizing activity against all of the SARS-CoV-2 variants tested (median titers ranging from 117 to 552) (Fig. 1c). Notably, titers remained within threefold of that observed for D614G for all variants except BA.4/5, which showed the highest degree of escape from serum-neutralizing antibodies (5.5-fold reduction from D614G), consistent with published serological studies[4,5]. Furthermore, the fold reduction in serum-neutralizing activity against SARS-CoV-2 VOCs relative to D614G remained similar at both time points, suggesting maintained serum neutralization breadth over time (Fig. 1d). We

observed minimal cross-neutralizing activity against SARS-CoV (median titer = 21) in all donors, suggesting that serum neutralization breadth remained limited to SARS-CoV-2 variants (Fig. 1c). We conclude that serum-neutralizing antibody responses remain at detectable levels against a diverse range of SARS-CoV-2 variants for up to 6-months following Omicron BA.1 breakthrough infection.

### BA.1 breakthrough infection induces cross-reactive memory B cell responses that persist for at least six months

Next, we assessed the magnitude and cross-reactivity of the antigen-specific B cell response via flow cytometric enumeration of B cells stained with differentially labeled wildtype (Wuhan-1; WT) and BA.1 RBD tetramers (Fig. 2a and Supplementary Fig. 1a). At the 5–6-month time point, total RBD-reactive B cells (WT and/or BA.1-reactive) and WT/BA.1 cross-reactive B cells comprised a median of 0.44% (ranging 0.12–2.53%) and 0.37% (ranging 0.12–2.53%) of class-switched (IgG+ or IgA+) B cells, respectively (Fig. 2b, c and Supplementary Fig. 1a). Thus, an average of 85% (ranging 69–100%) of all RBD+ class-switched B cells displayed BA.1/WT cross-reactivity at this time point, compared with 74% at 1-month post-infection (ranging 65–81%) (Fig. 2d and Supplementary Fig. 1b). Correspondingly, WT-specific B cells decreased from 26% of all RBD+ class-switched B cells at 1 month to 11% at 5–6 months (Fig. 2d and Supplementary Fig. 1b). Consistent with the waning of serum-neutralizing titers over time, we also observed a modest but statistically significant decline (1.1 to 3.7-fold) in the total frequencies of WT/BA.1 cross-reactive B cells between one- and 5–6-months following breakthrough infection (Fig. 2c). This result contrasts with that observed following primary SARS-CoV-2 infection and vaccination, where frequencies of spike-specific B cells have been shown to increase over the course of several months[18–21]. The reasons for this discrepancy are unclear but may be due to the increased magnitude of the initial short-lived B cell response and/or reduced germinal center size or longevity following secondary viral exposure. At the late time point, we also detected the emergence of a BA.1-specific B cell population in 3 of the 6 individuals (ranging from 1–18% of class-switched RBD+ B cells) (Fig. 2d and Supplementary Fig. 1b). In summary, Omicron BA.1 breakthrough infection induces a robust WT/BA.1 cross-reactive B cell response at early time points following infection, and this response only modestly declines over the course of 6 months.

### RBD-directed cross-reactive B cells display increased BA.1 binding affinity and neutralization potency over time

To compare the molecular characteristics of antibodies isolated at early and late time points following BA.1 breakthrough infection, we single-cell sorted 71 to 110 class-switched RBD-reactive B cells from four of the five previously studied donors (donors IML4042, IML4043, IML4044, and IML4045) at 139 to 170 days after breakthrough infection and expressed a total of 363 natively paired antibodies as full-length IgGs (Supplementary Fig. 1c)[12]. Antibodies isolated from the 5–6-month time point exhibited a high degree of clonal diversity, with four to 30% of antibodies belonging to clonal lineages shared with those identified at the early time point (Supplementary Fig. 3). Similar to the previously characterized antibodies from the acute time point, most of the newly isolated antibodies recognized both WT and BA.1 RBD antigens (73–97%), and we observed a bias toward certain VH germline genes (IGHV1–46, 1–69, 3–13, 3–53, 3–66, 3–9, and 4–31) (Fig. 2e and Supplementary Figs. 1d, 2a). Additionally, the level of SHM in the cross-reactive antibodies increased from a median of nine VH nucleotide substitutions at 1 month to 11 VH nucleotide substitutions by 5–6 months, potentially suggesting that BA.1 breakthrough infection drives further affinity maturation of pre-existing cross-reactive memory B cells (Fig. 2f).

Consistent with their higher levels of SHM, the antibodies isolated at 5–6 months displayed overall higher binding affinities for BA.1 (median $K_D$ = 2.2 nM and 1.3 nM at early and late time points,

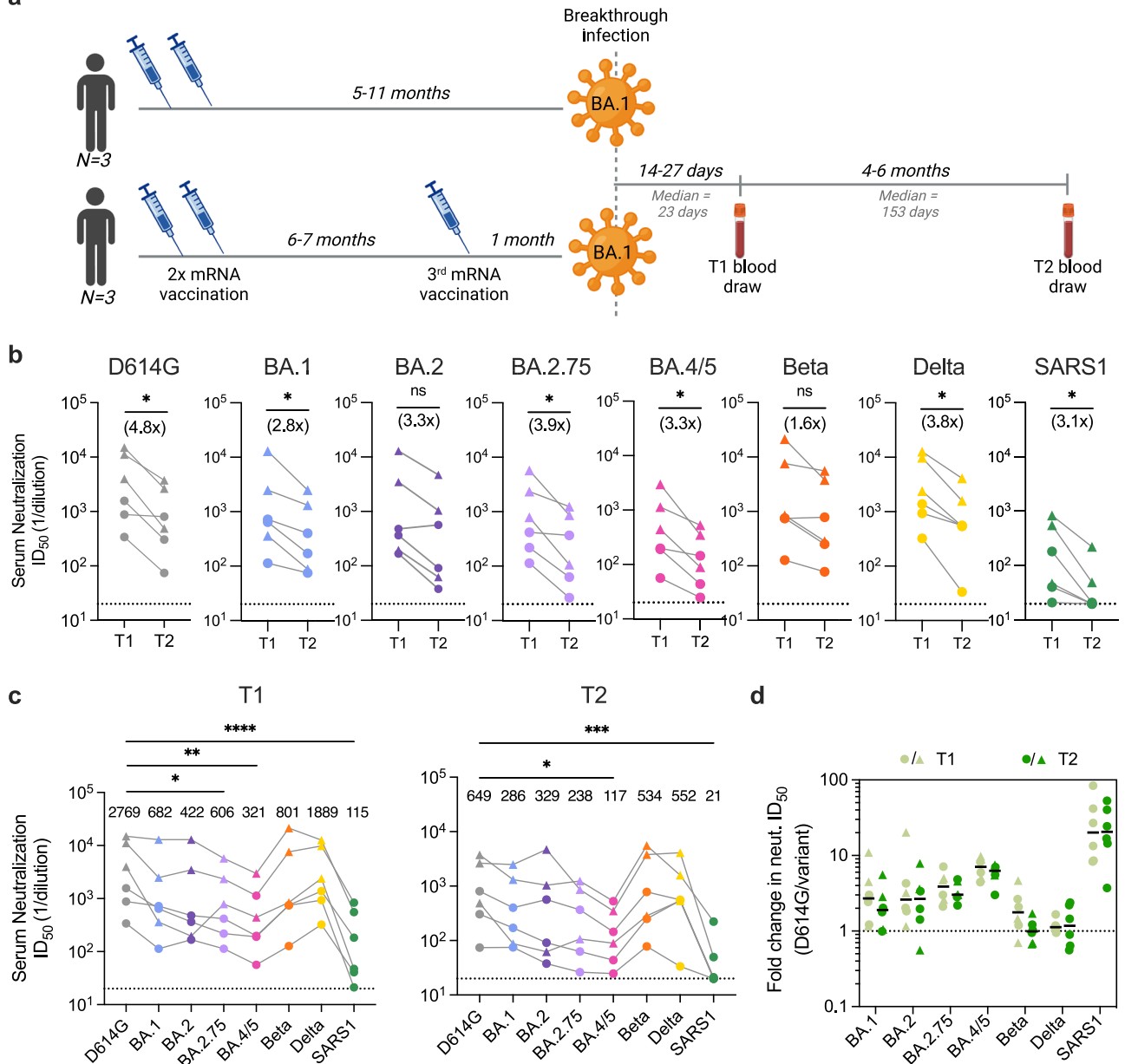

**Fig. 1 | Serum-neutralizing antibody responses at 1 month and 5–6 months following BA.1 breakthrough infection. a** Timeline of vaccination, BA.1 break-through infection, and sample collections. **b** Paired analysis of serum-neutralizing activity against SARS-CoV-2 D614G and BA.1, BA.2, BA.2.75, BA.4/5, Beta, and Delta variants, and SARS-CoV (SARS1) at 1-month[12] (T1) and 5–6-month (T2) time points, as determined using an MLV-based pseudovirus neutralization assay. Connected data points represent paired samples for each donor ($n = 6$ individuals), and the median fold change in serum titer between the two time points is shown in par-entheses. The dotted lines represent the lower limit of detection of the assay. **c** Serum neutralizing antibody titers against SARS-CoV-2 variants and SARS-CoV in samples collected at (left) 1-month[12] and (right) 5–6-month post-breakthrough infection for each donor ($n = 6$ individuals). Median titers are shown above the data points. The dotted lines represent the lower limit of detection of the assay. **d** Fold change in serum-neutralizing titers for the indicated SARS-CoV-2 variants and SARS-CoV relative to SARS-CoV-2 D614G at early[12] (T1) and late (T2) time points. Black bars represent median fold changes. The dotted line indicates no change in $ID_{50}$. Breakthrough infection donors infected after two-dose mRNA vaccination ($n = 3$) are shown as circles and those infected after a third mRNA dose ($n = 3$) are shown as triangles. Results are representative of two independent experiments. Statistical comparisons were determined by **b** two-tailed Wilcoxon matched-pairs signed rank test, **c** Friedman's one-way ANOVA with Dunn's multiple comparisons, or **d** two-way mixed model ANOVA. $ID_{50}$, 50% inhibitory dilution; *$P < 0.05$; **$P < 0.01$; ***$P < 0.001$; ****$P < 0.0001$; ns not significant. Source data and full sta-tistical test results are provided as a Source Data file.

respectively) and reduced binding to the WT RBD (median $K_D = 0.5$ and 1.0 nM at early and late time points, respectively) relative to early antibodies, suggesting that some antibodies suffered a loss in WT affinity during the process of affinity maturation toward BA.1 binding (Fig. 3a, b). These changes in binding recognition resulted in the late antibodies showing more balanced binding affinity profiles compared with the early antibodies (Fig. 3b). While the vast majority (97%) of early antibodies displayed a bias toward a higher binding affinity for

the WT RBD, nearly 50% of antibodies isolated at 5–6-months post-infection demonstrated a higher binding affinity for the BA.1 RBD (Fig. 3c). Furthermore, 73% of antibodies isolated at the late time point exhibited WT and BA.1 RBD affinities within twofold of each other compared to only 24% of early antibodies (Fig. 3c). Given the overall improvement in binding to the BA.1 RBD over time, we sought to explore whether the cross-reactive antibodies isolated at the late time point represented vaccine-induced clones that further matured

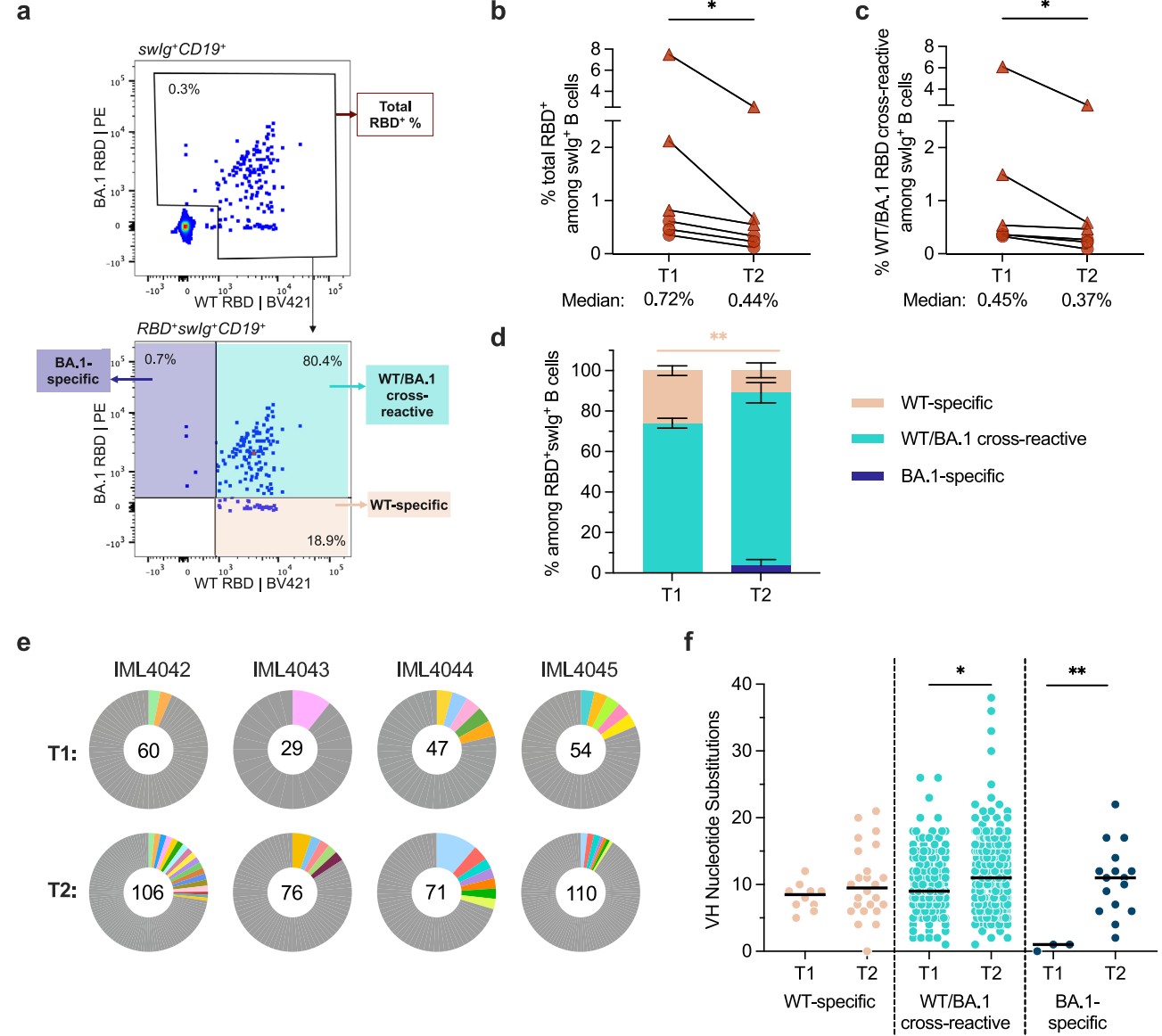

**Fig. 2 | Breakthrough infection induces durable SARS-CoV-2 RBD-specific memory B cell responses up to 6 months post-infection. a** Representative fluorescence-activated cell sorting gating strategy used to enumerate frequencies of (top) total (WT + BA.1) RBD-reactive B cells among class-switched (IgG[+] or IgA[+]) CD19[+] B cells and (bottom) WT-specific, BA.1-specific, and WT/BA.1 cross-reactive B cells among total RBD-reactive, class-switched (IgG[+] or IgA[+]) CD19[+] B cells. **b**, **c** Frequencies of **b** total RBD-reactive (*P* = 0.031) or **c** WT/BA.1 RBD cross-reactive (*P* = 0.032) B cells among class-switched CD19[+] B cells at 1-month[12] (T1) and 5–6-month (T2) time points. Connected data points represent paired samples for each donor. Donors infected after two-dose mRNA vaccination (*n* = 3) are shown as circles and those infected after a third mRNA dose (*n* = 3) are shown as triangles. **d** Mean proportions of RBD-reactive, class-switched B cells that display WT-specific, BA.1-specific, or WT/BA.1 cross-reactive binding at each time point (*n* = 6 donors, *P* = 0.015). Proportions were derived from 36–417 RBD-specific B cells analyzed per donor. Error bars indicate the standard error of mean. Statistical significance is shown for WT-specific antibodies; differences in the proportions of cross-reactive and BA.1-specific antibodies were non-significant. **e** Clonal lineage analysis of RBD-directed antibodies isolated from four donors at the early[12] (T1) and late (T2) time

points. Clonally expanded lineages (defined as antibodies with the same heavy and light chain germlines, same CDR3 lengths, and ≥80% CDRH3 sequence identity) are represented as colored slices. Each colored slice represents a clonal lineage, with the size of the slice proportional to the lineage size. Unique clones are combined into a single gray segment. The number of antibodies is shown in the center of each pie. Three of the donors (IML4042, IML4043, and IML4044) experienced BA.1 breakthrough infection following two-dose mRNA vaccination and the remaining donor (IML4045) was infected after a booster immunization. **f** Levels of somatic hypermutation, as determined by the number of nucleotide substitutions in the variable heavy (VH) region, at the early[12] (T1) and late (T2) time points among WT-specific (*n* = 146 and 283 at T1 and T2, respectively), WT/BA.1 cross-reactive (*n* = 10 and 24 at T1 and T2, respectively; *P* = 0.014), and BA.1-specific antibodies (*n* = 3 and 16 at T1 and T2, respectively; *P* = 0.002). Medians are shown by black bars. Statistical significance was determined by (**b**, **c**) two-tailed Wilcoxon matched-pairs signed rank test or (**d**, **f**) two-tailed Mann–Whitney *U*-test. swIg[+], class-switched immunoglobulin. **P* < 0.05; ***P* < 0.01. Source data and full statistical test results are provided as a Source Data file.

toward BA.1 following breakthrough infection or whether they originated from a de novo Omicron BA.1-induced B cell response. To investigate this question, we randomly selected ten BA.1-preferring antibodies from the late time point and produced their unmutated common ancestors (UCAs) as recombinant IgGs. Nine of the ten UCA

antibodies exhibited higher affinity binding to the WT RBD compared with the BA.1 RBD, suggesting that these antibodies likely originated from pre-existing vaccine-induced memory B cells that further affinity matured toward BA.1 following BA.1 breakthrough infection (Supplementary Fig. 4).

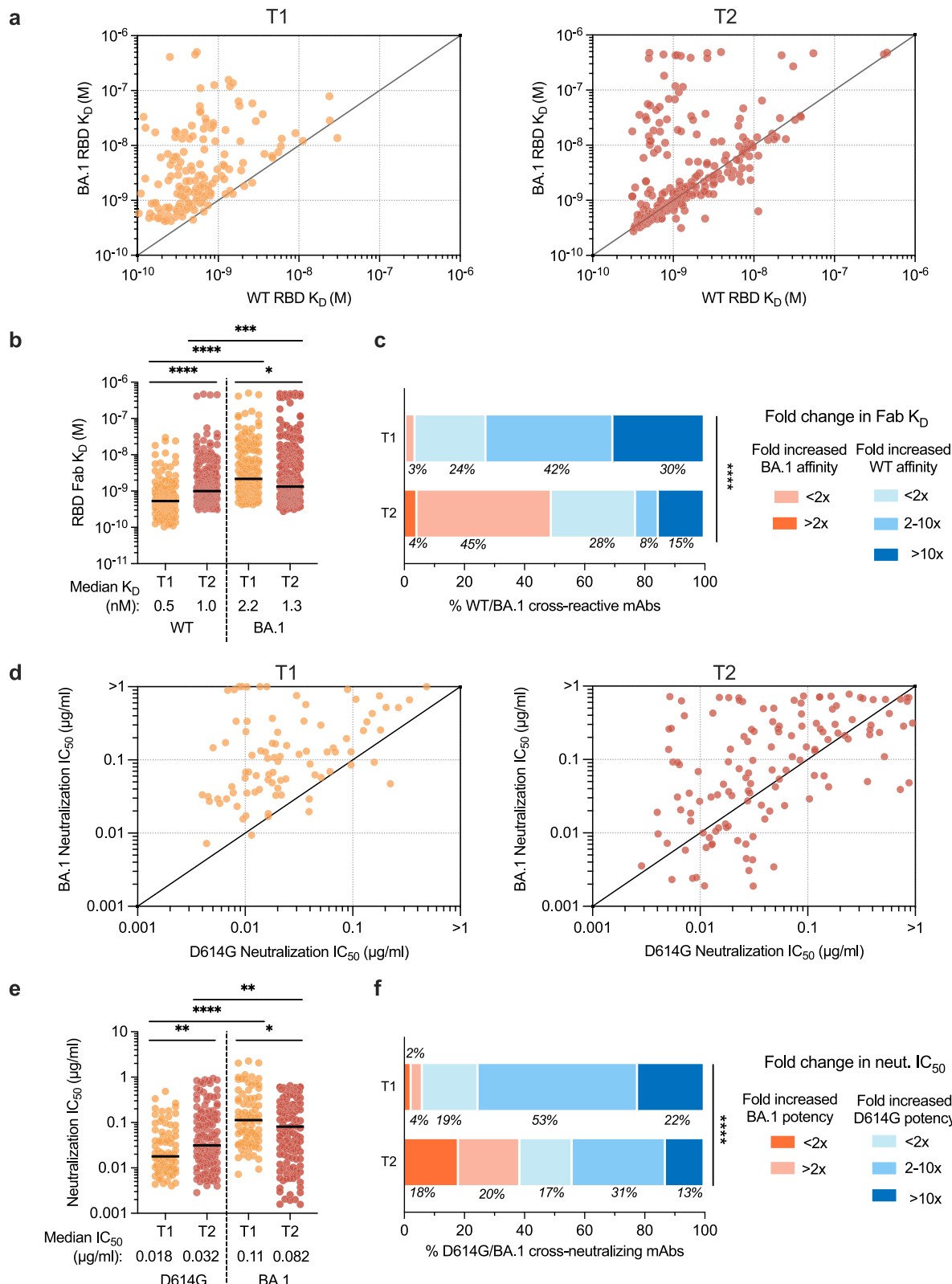

To determine whether the improvement in binding affinity for BA.1 translated into enhanced neutralization potency, we assessed the antibodies for neutralizing activity against WT and BA.1 using a pseudovirus assay. Fifty-one percent and 42% of WT/BA.1 cross-binding antibodies isolated from the 1-month and 5–6-month time point, respectively, displayed cross-neutralizing activity against D614G and BA.1 (defined as having $IC_{50}s < 2\,\mu g/ml$). At both time points studied,

neutralizing and non-neutralizing antibodies displayed similar SHM loads (Supplementary Fig. 5). Overall, the cross-neutralizing antibodies displayed approximately 2-fold lower potency against D614G at the late time point relative to the acute time point, consistent with the observed reduction in WT RBD affinity over time (Fig. 3d, e). As expected, the improvement in BA.1 binding affinities at the 5–6-month time point translated into an overall improvement in neutralization

**Fig. 3 | RBD-directed antibodies evolve toward enhanced binding and neutralizing activity. a**, **b** Fab binding affinities of WT/BA.1 cross-reactive antibodies for recombinant WT and BA.1 RBD antigens, as measured by BLI, are plotted as bivariates for antibodies derived from 1-month[12] (left, $n = 164$) and 5−6-month (right, $n = 280$) time points in (**a**) and summarized as a column dot plot in (**b**). Median affinities are indicated by black bars and shown below data points. **c** Proportions of WT/BA.1 cross-reactive antibodies at each time point that show an increased affinity for the BA.1 RBD relative to WT (red shades) or increased affinity for WT RBD (blue shades). Values represent the percentage of antibodies belonging to each of the indicated categories. **d**, **e** Potency of antibodies with cross-neutralizing activity against SARS-CoV-2 D614G and BA.1 (neutralization threshold defined as $IC_{50} < 2 \mu g/ml$), as determined by an MLV-based pseudovirus neutralization assay. $IC_{50}$ values are plotted in (**d**) as bivariates for antibodies isolated

from 1-month[12] (left, $n = 86$) and 5−6-month (right, $n = 132$) time points and summarized as column dot plots in (**e**). Median $IC_{50}$ values are indicated by black bars and shown below data points. **f** Proportions of WT/BA.1 cross-neutralizing antibodies at each time point that show increased neutralizing potency against BA.1 (red shades) or D614G (blue shades). Values represent the percentage of antibodies belonging to each of the indicated categories. Statistical comparisons were determined by (**b**, **e**) multiple two-tailed Mann−Whitney $U$−tests without adjustment for multiplicity across time points and two-tailed Wilcoxon matched-pairs rank tests within each time point or (**c**, **f**) two-tailed Mann−Whitney $U$-test. $IC_{50}$ 50% inhibitory concentration, $K_D$ equilibrium dissociation constant; *$P < 0.05$; **$P < 0.01$; ***$P < 0.001$; ****$P < 0.0001$. Source data and full statistical test results are provided as a Source Data file.

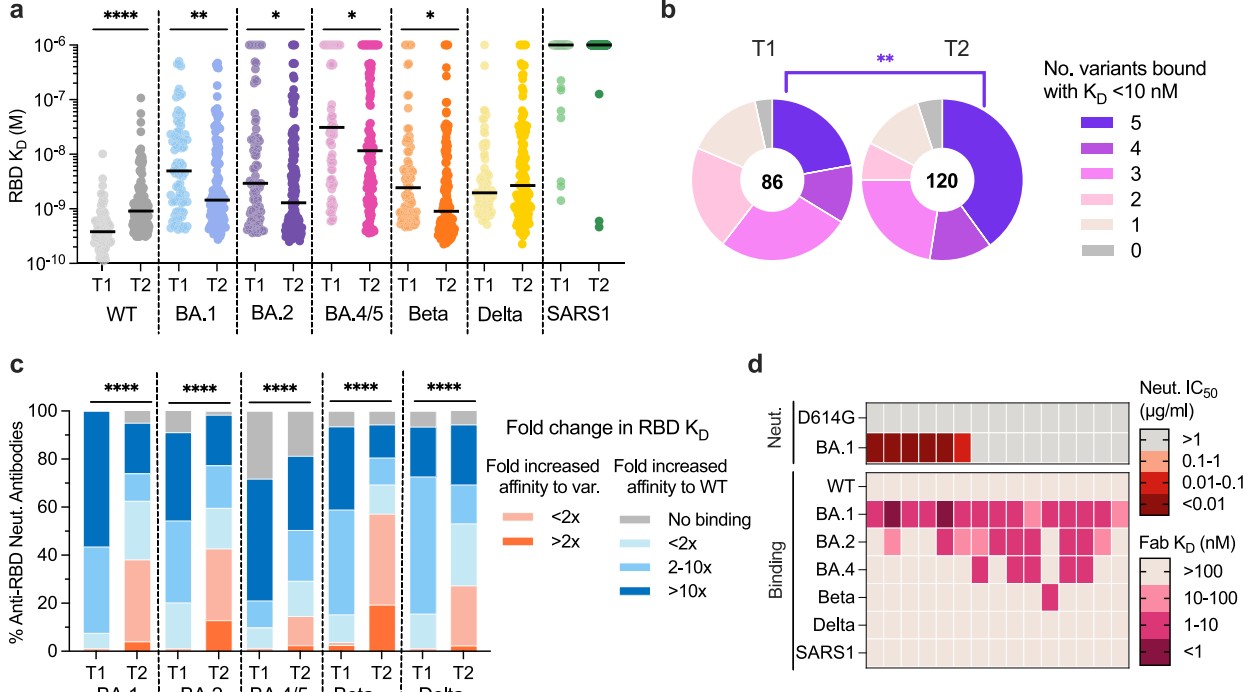

**Fig. 4 | Improved breadth of activity of D614G/BA.1 cross-neutralizing antibodies over time. a** Fab binding affinities of D614G/BA.1 cross-neutralizing antibodies isolated at 1-month[12] (T1, $n = 86$) and 5−6-month (T2, $n = 132$) time points for recombinant SARS-CoV-2 variant RBDs and the SARS-CoV RBD, as determined by BLI. Black bars represent medians. **b** Pie charts showing the proportions of antibodies derived from (left) early[12] and (right) late time points that bound the indicated number of SARS-CoV-2 variant RBDs with Fab $K_D$s < 10 nM (statistical $P = 0.026$). The total number of antibodies is shown in the center of each pie. **c** Proportions of D614G/BA.1 cross-neutralizing antibodies from early[12] ($n = 86$) and

late ($n = 132$) time points with the indicated fold changes in Fab binding affinities for recombinant SARS-CoV-2 variant RBDs relative to the WT RBD. **d** Heatmap showing neutralization $IC_{50}$s and SARS-CoV-2 variant RBD binding affinities of BA.1-specific antibodies. Statistical comparisons were determined by **a**, **c** Kruskal−Wallis test with Holms corrected multiple pairwise comparisons or **b** two-sided Fisher's exact test. $IC_{50}$ 50% inhibitory concentration, $K_D$ equilibrium dissociation constant; *$P < 0.05$; **$P < 0.01$; ****$P < 0.0001$. Source data and full statistical test results are provided as a Source Data file.

potency (Fig. 3e). Approximately 16% of antibodies isolated at 5−6 months displayed neutralization $IC_{50}$s < 0.01 ug/ml against BA.1 compared to only 2% of antibodies isolated at the earlier time point (Supplementary Fig. 6). Correspondingly, 38% of the neutralizing antibodies isolated at 5−6 months exhibited more potent activity against BA.1 relative to D614G, compared to only 6% of the neutralizing antibodies isolated at the early time point (Fig. 3f). In summary, WT/BA.1 cross-reactive antibodies induced following BA.1 breakthrough infection evolve toward increased BA.1 affinity and neutralization potency for at least 6 months post-infection.

## WT/BA.1 cross-neutralizing antibodies evolve enhanced breadth of activity against SARS-CoV-2 variants over time

To evaluate the breadth of WT/BA.1 cross-neutralizing antibodies induced following BA.1 breakthrough infection, we evaluated

their binding reactivities with a panel of recombinant RBDs encoding mutations present in SARS-CoV-2 variants BA.2, BA.4/5, Beta, and Delta, and the more antigenically divergent SARS-CoV. D614G/BA.1 cross-neutralizing antibodies isolated at the late time point displayed a 2.4-fold reduced affinity for the WT RBD and 3.4-fold improved affinity for the BA.1 RBD relative to early neutralizing antibodies, consistent with the pattern observed for all WT/BA.1 cross-binding antibodies (Figs. 4a, 3a−c). Furthermore, the WT/BA.1 cross-reactive antibodies isolated at 5−6 months broadly recognized other SARS-CoV-2 variants, except for BA.4/5, for which we observed a ≥5-fold loss in affinity for 57% (68/120) of the WT/BA.1 neutralizing antibodies (Fig. 4a and Supplementary Fig. 7). Importantly, the 5−6-month antibodies displayed higher affinity binding to all Omicron subvariants and Beta relative to the early antibodies, suggesting that the observed affinity maturation toward BA.1 also translated into the improved breadth of reactivity

with other variants (Fig. 4a). In support of this finding, a significantly higher proportion (40%) of neutralizing antibodies isolated at 5–6 months displayed high affinity ($K_D < 10$ nM) binding to all five variants tested compared to the neutralizing antibodies isolated at the acute time point (22%) (Fig. 4b). Furthermore, antibodies isolated at the late time point displayed more balanced binding affinity profiles for BA.1, BA.2, BA.4/5, and pre-Omicron variants (Beta and Delta) relative to antibodies isolated from the acute time point (Fig. 4c). We conclude that BA.1 breakthrough infection results in an overall broadening of the anti-RBD neutralizing antibody repertoire.

### BA.1 breakthrough infection induces limited BA.1-specific antibody responses with a narrow breadth of activity

Although the vast majority of antibodies isolated at the 5–6-month time point displayed WT/BA.1 cross-reactive binding, we identified a limited number of BA.1-specific antibodies (2 to 8 per donor) in all four donors, comprising 1 to 15% (median = 4%) of total RBD-specific antibodies (Supplementary Fig. 1d). In contrast, we only detected BA.1-specific antibodies in a single donor at the acute time point (Supplementary Fig. 1d). Furthermore, the BA.1-specific antibodies identified at 5–6 months displayed relatively high levels of SHM (median = 11 VH nucleotide substitutions), similar to those of WT/BA.1 cross-reactive antibodies, suggesting that BA.1-specific antibodies had undergone the process of affinity maturation in GCs (Fig. 2f). Six of 15 (40%) BA.1-specific antibodies isolated at the late time point neutralized BA.1, with $IC_{50}$s ranging from 0.002 to 0.089 µg/ml, and none of these antibodies displayed detectable neutralizing activity against D614G (Fig. 4d). In contrast to the broad binding activity exhibited by the WT/BA.1 cross-reactive antibodies, the BA.1-specific antibodies displayed limited breadth, with only half of neutralizing antibodies maintaining binding to BA.2 and none of the antibodies showing reactivity with WT, BA.4/5, Beta, or Delta (Fig. 4d). To investigate whether the BA.1-specific antibodies originated from a de novo B cell response or pre-existing vaccine-induced memory B cells that lost WT reactivity during the process of affinity maturation toward BA.1, we generated UCAs from four clonally distinct BA.1-specific antibodies and measured their binding affinities to WT and BA.1 RBDs. All four UCAs showed reduced binding to the WT RBD relative to BA.1, with three of the four UCAs displaying no detectable WT recognition, suggesting that these B cells likely represent a de novo response induced by BA.1 breakthrough infection (Supplementary Fig. 4). Thus, BA.1 breakthrough infection induces a limited and delayed de novo Omicron-specific B cell response that affinity matures over time.

### Convergent clones dominate the neutralizing antibody response at both early and late time points and their escape mutations predict Omicron evolution

Among D614G/BA.1 cross-neutralizing antibodies isolated at both time points, we observed significant over-representation of four IGHV germline genes (*IGHV1–69, IGHV3–53/3–66*, and *IGHV3–9*)[12] (Supplementary Fig. 2b). At the 5–6-month time point, over half (54%) of the neutralizing antibodies were encoded by one of these four germline genes, with one-third of these antibodies utilizing *IGHV1–69* (Fig. 5a and Supplementary Fig. 8a). We previously reported that BA.1-neutralizing *IGHV1–69* antibodies isolated at the acute time point preferentially paired with the light chain germline IGLV1–40 and targeted an antigenic site overlapping that of the class 3 antibody COV2-2130 and non-overlapping with the ACE2 binding site[12]. Similarly, 69% of *IGHV1–69* antibodies isolated at the 5–6-month time point paired with *IGLV1–40* light chains and the majority (80%) failed to compete with ACE2 for binding (Supplementary Figs 8b, 7c). Likewise, >90% of *IGHV3–9* antibodies identified from both time points recognized epitopes outside of the ACE2 binding site (Supplementary Fig. 8c). However, in contrast to *IGHV1–69* antibodies, most of the *IGHV3–9* antibodies competed with the class 3 antibodies S309 and REGN10987

for binding (in addition to COV2-2130), suggesting a distinct mode of recognition[12]. Lastly, BA.1-neutralizing *IGHV3–53/66* antibodies isolated from both time points were characterized by short HCDR3s (median = 11 to 12 amino acids) and displayed competitive binding with the ACE2 receptor (Fig. 5b and Supplementary Fig. 8c). Thus, convergent antibody classes dominated the neutralizing antibody response at both early and late time points following BA.1 breakthrough infection, suggesting that B cell affinity maturation did not dramatically impact RBD epitope immunodominance hierarchy.

Given the well-established role of convergent classes of neutralizing antibodies in shaping SARS-CoV-2 evolution, combined with the unprecedented magnitude of the Omicron BA.1 infection wave, we sought to map the mutations associated with Omicron BA.1 escape from the four common classes of neutralizing antibodies induced following breakthrough infection[18,22]. We randomly selected one to two antibodies belonging to each convergent germline and performed deep mutational scanning (DMS) analysis using a library encoding all possible amino acid substitutions from BA.1 (Supplementary Fig. 8d–e)[23]. Antibodies encoded by *IGHV3–53* (ADI-75733) and *IGHV3–66* (ADI-75732) displayed similar escape profiles, consistent with their shared sequence features and competitive binding profiles (Fig. 5c–d)[12]. RBD positions N460 and F486, which are mutated in newly emergent SARS-CoV-2 variants (e.g., N460K in B.2.75, BA.2.75.2, BN.1, BQ.1, and XBB; F486S in BA.2.75.2 and XBB; and F486V in BA.4/5, BA.4.6, and BQ.1.1), were associated with binding escape from *IGHV3–53/66* antibodies (Fig. 5c–e). *IGHV1–69* and *IGHV3–9* antibodies both showed reduced binding to RBDs incorporating mutations at positions 344–349, 356, 452–453, 468, and 490. Notably, residues R346, K356, L452, and F490 are mutated across evolutionarily diverse Omicron sublineages, including BA.4.6 (R346T, L452R), BA.4/5 (L452R), BA.2.12.1 (L452Q), BJ.1 (R346T, F490V), BN.1 (R346T, K356T, F490S), and BQ.1.1 (R346T, L452R), and XBB (R346T, F490S) (Fig. 5c–e). Consistent with these escape profiles, *IGHV1–69* and *IGHV3–9* class antibodies displayed reduced binding to BA.2.12.1 and BA.4/5 relative to early Omicron variants, likely due to the unique L452Q/R mutations present in these variants compared with BA.1 and BA.2 (Fig. 5f). Consistent with DMS-based predictions, both BA.2.75 and BA.4/5 RBDs displayed increased binding resistance to *IGHV3–53/66* antibodies (Fig. 5e, f). Thus, convergent D614G/BA.1 cross-neutralizing antibodies recognize epitopes commonly mutated in recently emerging Omicron subvariants, providing a molecular explanation for the high degree of antigenic convergence observed in recent Omicron subvariants and their increased level of immune evasion relative to BA.1.

## Discussion

In conclusion, Omicron BA.1 breakthrough infection in mRNA-vaccinated individuals induces broadly neutralizing serological and MBC responses that persist at detectable levels for at least 6 months following infection, supporting real-world studies showing that BA.1 breakthrough infection provides protection against symptomatic BA.1, BA.2, and BA.5 infection for at least 5–6 months[24–26]. Furthermore, although the acute B cell response is primarily mediated by the re-activation of cross-reactive MBCs originally induced by vaccination, a subset of vaccine-induced MBC clones accumulates somatic mutations and evolves increased breadth and potency for at least 5–6 months following BA.1 breakthrough infection. Unfortunately, due to the lack of a comparison cohort comprised of unvaccinated individuals experiencing primary BA.1 infection, it was not possible to precisely determine the proportion of cross-reactive clones originating from de novo versus pre-existing vaccine-induced memory B cell responses. However, the observation that inferred germline ancestors of BA.1-preferring, cross-reactive antibodies display biased recognition of the WT RBD suggests that a large proportion of such clones were originally induced by the ancestral vaccine strain and subsequently affinity matured toward BA.1 following breakthrough infection. Future

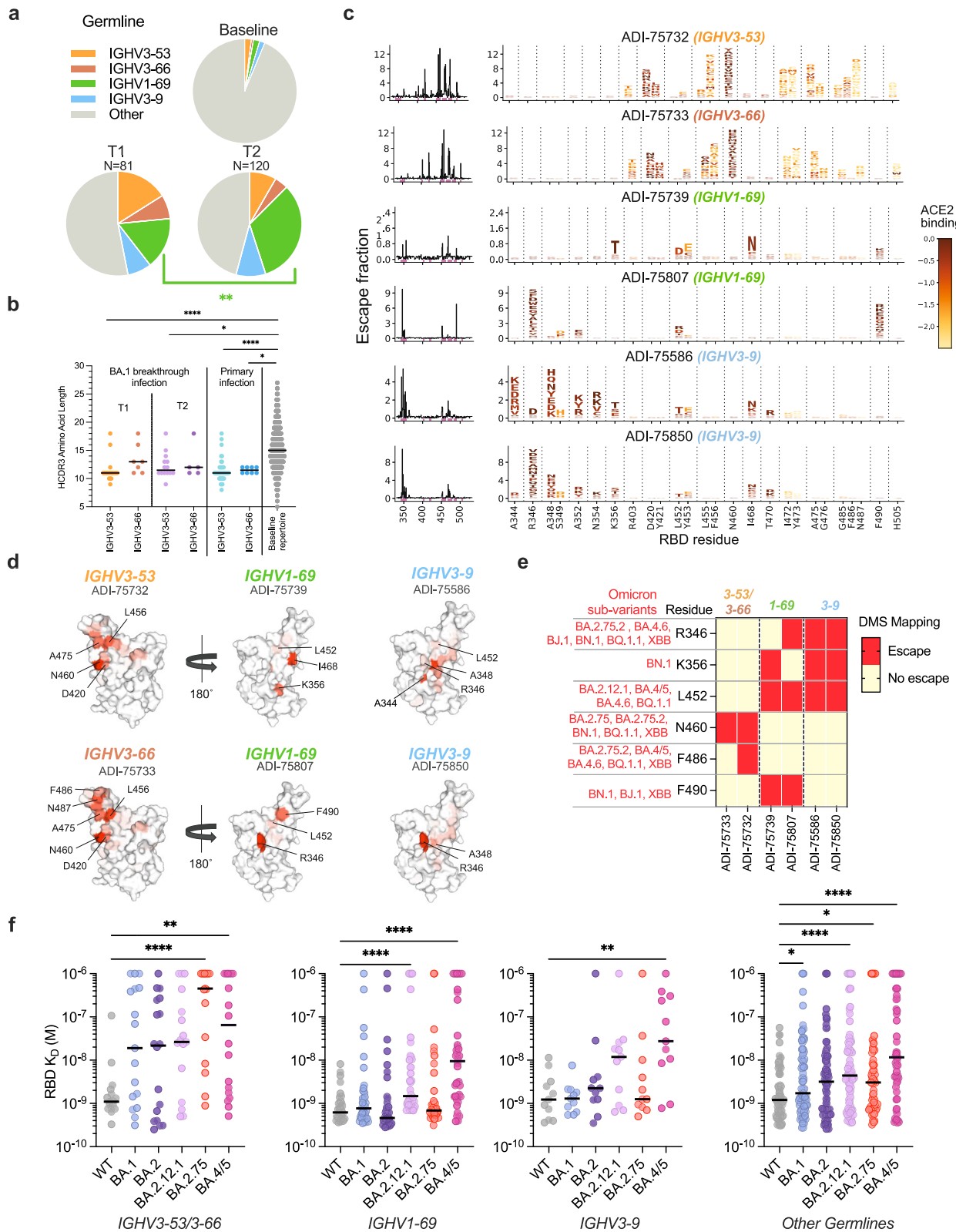

longitudinal studies analyzing serum and memory B cell responses following primary BA.1 infection in unvaccinated individuals or direct probing of germinal center B cells at early and late time points following breakthrough infection may provide more comprehensive answers to this question. Although the enhanced neutralization breadth and potency observed in the memory B cell compartment at 5–6 months post-infection was not reflected in the serum antibody

response, it is possible that a second heterologous exposure may broaden the serological repertoire by activating these newly affinity matured MBCs, akin to the improved serum neutralization breadth and potency observed following a third mRNA vaccine dose[21,27]. Nevertheless, our data indicate that infection or vaccination with antigenically divergent SARS-CoV-2 variants may provide long-term benefits by broadening pre-existing anti-SARS-CoV-2 B cell memory.

**Fig. 5 | BA.1 neutralizing antibodies display convergent sequence and binding properties. a** Pie charts showing frequencies of the indicated convergent germline genes among D614G/BA.1 cross-neutralizing antibodies isolated at early[12] (T1) and late (T2) timelines (statistical $P = 0.0021$). Germline gene frequencies observed in baseline human antibody repertoires (upper right) are shown for comparison[39]. **b** HCDR3 amino acid length distribution of *IGHV3–53* and *IGHV3–66* cross-neutralizing antibodies isolated 1-month[12] (T1, $n = 28$) and 5–6 months (T2, $n = 19$) following BA.1 breakthrough infection. HCDR3 lengths of *IGHV3–53/3–66*-utilizing antibodies isolated following primary D614G infection ($n = 51$) and the baseline human antibody repertoire ($n = 30,546$) were included for comparison[18,39]. **c** Line plots at left show the total site-wise escape at each RBD site, as determined using deep mutational scanning analysis of yeast-displayed SARS-CoV-2 BA.1 RBD mutant libraries. Sites of strong escape indicated by pink bars are shown at the mutation level in logo plots. Mutations are colored by their effects on ACE2 binding (scale bar

at right). **d** Structural projections of binding escape mutations determined for the indicated convergent antibodies. The RBD surface is colored by a gradient ranging from no escape (white) to strong escape (red) at each site. See Supplementary Fig. 8 for additional details. **e** Heatmap summarizing convergent antibody-escape mutations present in the indicated SARS-CoV-2 Omicron sublineages. **f** Fab binding affinities of convergent antibodies utilizing the indicated germline genes ($n = 18$ for *IGHV3–53/66*, $n = 40$ for *IGHV1–69*, $n = 11$ for *IGHV3–9*, and $n = 58$ for other germlines) for SARS-CoV-2 WT and Omicron sub-variant RBD antigens, as measured by BLI. Black bars indicate median affinities. Statistical comparisons were determined by **a** two-sided Fisher's exact test or **b, f** Kruskal–Wallis test with subsequent Dunn's multiple comparisons with WT. HCDR3 heavy chain complementarity-determining region 3, $K_D$ equilibrium dissociation constant; $*P < 0.05$; $**P < 0.01$; $****P < 0.0001$. Source data and full statistical test results are provided as a Source Data file.

At both time points studied, we detected few to no BA.1-reactive antibodies that lacked WT cross-reactivity, suggesting that BA.1 breakthrough infection induces limited BA.1-specific de novo B cell responses in individuals with pre-existing immunity. Consistent with these results, booster immunization with a monovalent Omicron BA.1-based mRNA vaccine (mRNA-1273.529) minimally induced de novo BA.1-specific B cell responses in humans[28]. However, it is possible that de novo Omicron-specific B cell responses may be further amplified following a secondary Omicron exposure, similar to that observed following H5N1 influenza virus immunization in humans[29].

Finally, we found that convergent classes of neutralizing antibodies dominated the Omicron BA.1 breakthrough response at both early and late time points, which is reminiscent of the public antibody response elicited following primary infection or vaccination with early ancestral SARS-CoV-2 strains[30–32]. The rapid emergence and global spread of Omicron subvariants with substitutions in the binding sites of these highly potent convergent clones provides strong evidence that these types of antibodies are applying the immune pressure driving the continued antigenic drift of Omicron. Thus, in contrast to current approaches to the design of universal vaccines for certain highly antigenically variable viruses, such as HIV and influenza, which aim to focus the neutralizing response on a limited number of relatively conserved epitopes, the development of "variant-proof" COVID-19 vaccines may require the development of spike-based immunogens that induce a diversity of neutralizing antibodies targeting numerous co-dominant epitopes, with the goal of limiting convergent immune pressure and therefore constraining viral evolution[33–35].

This study is subject to several potential limitations. First, we studied a relatively small number of donors, all of whom were young, Caucasian, and experienced mild disease. Rather than study a large cohort with few antibodies isolated from each individual, we chose to conduct in-depth characterization on a large number of monoclonal antibodies isolated from a limited set of donors at both early and late time points following breakthrough infection. Moreover, there is variability among donors in terms of the number of mRNA vaccine doses received (either two or three doses) prior to breakthrough infection, the length of time between vaccination and infection, and the timing of sample collection after infection. As such, observations within donors can be made with high certainty, but given the relatively small cohort size, demographic biases, and variability in exposure history, caution should be exercised in the generalization of these results to the broader population. Further, due to the unpredictability of natural infection, we were unable to collect serum samples from these donors at a time point prior to breakthrough infection, which precluded comparisons with the pre-infection vaccine-induced repertoires. In the absence of such samples, it is not possible to definitively determine whether Omicron breakthrough infection activated pre-existing vaccine-induced MBCs. Finally, due to limitations in serum sample availability, we were unable to perform serum depletion studies to determine the relative contributions

of de novo BA.1-specific versus cross-reactive antibodies to the serum antibody response following BA.1 breakthrough infection.

## Methods

### Human subjects and blood sample collection

Seven BA.1 breakthrough-infected participants were recruited to participate in this study with informed consent under the Dartmouth Health Protocol D10083, approved by the Dartmouth Health Human Research Protection Program Institutional Review Board, and overseen by DartLab, a Dartmouth Cancer Center Shared Resource. Participants were compensated $3 for every 10 ml of blood given. Sex and gender were not considered due to limitations in sample availability and the exploratory nature of the study design. Briefly, participants experienced breakthrough infection after two- or three-dose mRNA vaccination (BNT162b2 and/or mRNA-1273). Venous blood was collected at two time points, an early visit at 14 to 27 days (T1) and a late visit at 139 to 170 days (T2) after their first SARS-CoV-2 test. Participants had no documented history of SARS-CoV-2 infection prior to vaccination or between the two blood draw time points. Clinical and demographic characteristics of breakthrough infection donors are shown in Supplementary Table 1 and Supplementary Table 2. Plasma and peripheral blood mononuclear cell (PBMC) samples were isolated using a Ficoll 1077 (Sigma) gradient in SepMate™ PBMC Isolation Tubes, following the manufacturer's recommendations. PBMCs were aliquoted to 10 million cells per vial and stored in liquid nitrogen. Isolated plasma was centrifuged at $1000 \times g$ for 10 min, and the supernatant was aliquoted and stored at −80 °C.

### Plasmid design and construction

Plasmids expressing spike proteins of SARS-CoV-2 variants and SARS-CoV were ordered as gene block fragments (IDT) and cloned into a mammalian expression vector (pcDNA3.3) for MLV-based pseudovirus production[36]. All SARS-CoV-2 variant spikes and the SARS-CoV spike were C-terminally truncated by 19-amino acids or 28-amino acids, respectively, to increase infectious titers. The SARS-CoV S sequence was retrieved from ENA (AAP13441). SARS-CoV-2 variants contain the following mutations from the Wuhan-Hu-1 sequence (Genbank: NC_045512.2):

- D614G: D614G
- Beta: D80A, D215G, Δ242–244, K417N, E484K, N501Y, D614G, A701V
- Delta: T19R, G142D, Δ156–157, R158G, L452R, T478K, D614G, P681R, D950N
- BA.1: A67V, Δ69–70, T95I, G142D/Δ143–145, Δ211/L212I, ins214EPE, G339D, S371L, S373P, S375F, K417N, N440K, G446S, S477N, T478K, E484A, Q493R, G496S, Q498R, N501Y, Y505H, T547K, D614G, H655Y, N679K, P681H, N764K, D796Y, N856K, Q954H, N969K, L981F
- BA.2: T19I, L24S, Δ25–27, G142D, V213G, G339D, S371F, S373P, S375F, T376A, D405N, R408S, K417N, N440K, S477N, T478K,

E484A, Q493R, Q498R, N501Y, Y505H, D614G, H655Y, N679K, P681H, N764K, D796Y, Q954H, N969K

- BA.4/5: T19I, L24S, Δ25−27, Δ69−70, G142D, V213G, G339D, S371F, S373P, S375F, T376A, D405N, R408S, K417N, N440K, L452R, S477N, T478K, E484A, F486V, Q498R, N501Y, Y505H, D614G, H655Y, N679K, P681H, N764K, D796Y, Q954H, N969K
- BA.2.75: T19I, L24S, Δ25−27, G142D, K147E, W152R, F157L, I210V, V213G, G339H, G257S, S371F, S373P, S375F, T376A, D405N, R408S, K417N, N440K, G446S, N460K, S477N, T478K, E484A, Q498R, N501Y, Y505H, D614G, H655Y, N679K, P681H, N764K, D796Y, Q954H, N969K

### SARS-CoV-2 pseudovirus generation
Single-cycle infectious MLVs were pseudotyped with the spike proteins of SARS-CoV-2 variants and SARS-CoV[36]. HEK293T cells (ATCC CRL-3216) were seeded at a density of 0.5 million cells/ml in six-well tissue culture plates and the next day, transfected using Lipofectamine 2000 (Thermo Fisher Scientific) with the following plasmids: (1) 0.5 μg per well of pCDNA3.3 encoding SARS-CoV-2 spike with a 19-amino acid truncation at the C-terminus, (2) 2 μg per well of MLV-based luciferase reporter gene plasmid (Vector Builder), and (3) 2 μg per well of of MLV gag/pol (Vector Builder). MLV particles were harvested 48 h post-transfection, aliquoted, and stored at −80 °C for neutralization assays.

### Pseudovirus neutralization assay
56 °C heat-inactivated sera or antibodies were serially diluted in 50 μl MEM/EBSS media supplemented with 10% fetal bovine serum (FBS) and incubated with 50 μl of MLV viral stock for 1 h at 37 °C. Following incubation, antibody-virus mixtures were added to previously seeded HeLa-hACE2 reporter cells (BPS Bioscience Cat #79958). Infection was allowed to occur for 48 h at 37 °C. Infection was measured by lysing cells with Luciferase Cell Culture Lysis reagent (Promega) and detecting luciferase activity using the Luciferase Assay System (Promega) following the manufacturer's protocols. Infectivity was quantified by relative luminescence units (RLUs) and the percentage neutralization was calculated as $100 \times (1 - [RLU_{sample} - RLU_{background}] / [RLU_{isotype\ control\ mAb} - RLU_{background}])$. Neutralization $IC_{50}$ was interpolated from curves fitted using four-parameter non-linear regression in GraphPad Prism (version 9.3.1). Reported results are representative of replicate experiments performed from multiple pseudovirus batches.

### FACS analysis of SARS-CoV-2 S-specific B cell responses
Antigen-specific B cells were detected using recombinant biotinylated antigens tetramerized with fluorophore-conjugated streptavidin (SA). Avitag biotinylated WT RBD (Acro Biosystems, Cat #SPD-C82E8) and Avitag biotinylated BA.1 RBD (Acro Biosystems, Cat # SPD-C82E4) were mixed in 4:1 molar ratios with SA-BV421 (BioLegend) and SA-phycoerythrin (PE; Invitrogen), respectively, and allowed to incubate for 20 min on ice. Unbound SA sites were subsequently quenched using 5 μl of 2 μM Pierce biotin (Thermo Fisher Scientific). Approximately 10 million PBMCs were stained with tetramerized RBDs (25 nM each); anti-human antibodies anti-CD19 (PE-Cy7; Clone HIB19; Biolegend Cat # 302216), anti-CD3 (PerCP-Cy5.5; Clone OKT3; Biolegend Cat # 317335), anti-CD8 (PerCP-Cy5.5; Clone SK1; Biolegend Cat # 344710), anti-CD14 (PerCP-Cy5.5; Clone 61D3; Invitrogen Cat # 45-0149-42), and anti-CD16 (PerCP-Cy5.5; Clone B73.1; Biolegend Cat # 360712); and 50 μl Brilliant Stain Buffer (BD BioSciences) diluted in FACS buffer (2% BSA/1 mM EDTA in 1X PBS). All antibodies were used at 1:100 dilutions. About 200 μl of staining reagents were added to each PBMC sample and incubated for 15 min on ice. After one wash with FACS buffer, cells were stained in a mixture of propidium iodide and anti-human antibodies anti-IgG (BV605; Clone G18−145; BD Biosciences Cat # 563246), anti-IgA (FITC; Abcam Cat # Ab98553), anti-CD27 (BV510; Clone M-T271; BD Biosciences Cat # 740167), and anti-CD71 (APC-Cy7; Clone

CY1G4; Biolegend Cat # 334110). Following 15 min of incubation on ice, cells were washed two times with FACS buffer and analyzed using a BD FACSAria II (BD BioSciences).

For sorting of RBD-specific, class-switched B cells, PBMCs that react with either WT and/or BA.1 RBD tetramers among CD19+CD3−CD8−CD14−CD16−PI− and IgG+ or IgA+ cells were single-cell index sorted into 96-well polystyrene microplates (Corning) containing 20 μl lysis buffer per well [5 μl of 5X first strand SSIV cDNA buffer (Invitrogen), 1.25 μl dithiothreitol (Invitrogen), 0.625 μl of NP-40 (Thermo Scientific), 0.25 μl RNaseOUT (Invitrogen), and 12.8 μl dH2O]. Plates were briefly centrifuged and then frozen at −80 °C before PCR amplification.

### Amplification and cloning of antibody variable genes
Antibody variable gene fragments (VH, Vk, and Vλ) were amplified by RT-PCR as described previously in ref. [37]. Briefly, cDNA was synthesized using randomized hexamers and SuperScript IV enzyme (Thermo Fisher Scientific). cDNA was subsequently amplified by two rounds of nested PCRs, with the second cycle of nested PCR adding 40 base pairs of flanking DNA homologous to restriction enzyme-digested *S. cerevisiae* expression vectors to enable homologous recombination during transformation. PCR-amplified variable gene DNA was mixed with expression vectors and chemically transformed into competent yeast cells via the lithium acetate method[38]. Yeast were plated on selective amino acid drop-out agar plates and individual yeast colonies were picked for sequencing and recombinant antibody expression.

### Expression and purification of IgG and Fab molecules
Antibodies were expressed as human IgG1 via *S. cerevisiae* cultures, as described previously[37]. Briefly, yeast cells were grown in culture for 6 days for antibody production, before collecting IgG-containing supernatant by centrifugation. IgGs were subsequently purified by protein A-affinity chromatography and eluted using 200 mM acetic acid/50 mM NaCl (pH 3.5). The pH was then neutralized using 1/8th volume of 2 M Hepes (pH 8.0). Fab fragments were cleaved from full-length IgG by incubating with papain for 2 h at 30 °C before terminating the reaction using iodoacetamide. Fab fragments were purified from the mixture of digested antibody Fab ad Fc fragments using a two-step chromatography system: (1) Protein A agarose was used to remove Fc fragments and undigested IgG and (2) Fabs in the flow-through were further purified using CaptureSelect™ IgG-CH1 affinity resin (Thermo Fisher Scientific) and eluted from the column using 200 mM acetic acid/50 mM NaCl (pH 3.5). Fab solutions were pH-neutralized using 1/8th volume 2 M Hepes (pH 8.0).

### Binding analysis of unmutated common ancestors by ELISA
96-well half-area plates (Corning) were coated with 25 μl of 5 μg/ml of recombinant His-tagged WT or BA.1 RBD diluted in PBS overnight at 4 °C. The next day, wells were blocked with 50 μl of 3% bovine serum albumin (BSA) in 1X PBS for 1 h at room temperature, and subsequently, washed two times using wash buffer (1X PBS, 0.05% Tween-20). Next, antibody titrations diluted in 1% BSA, 0.05% Tween-20 in 1X PBS were added to plates and incubated for 1 h at 37 °C before washing three times with wash buffer. Antigen-bound antibodies were detected using alkaline phosphatase-conjugated anti-human IgG (Jackson ImmunoResearch, Cat #109-055-098) diluted 1:1000 in 1% BSA, 0.05% Tween-20 in 1X PBS for 1 h at 37 °C. Plates were then washed three times and developed with 25 μl of alkaline phosphatase staining buffer (pH 9.8) for 15 min. Absorbance was measured at 405 nm using a spectrophotometer (VersaMax). Experiments were performed in duplicate using the same IgG preparations, and the area under the curve was calculated after subtracting background absorbance from a no-antibody control.

### Binding affinity measurements by biolayer interferometry
Antibody binding kinetics were measured by biolayer interferometry (BLI) using a FortéBio Octet HTX instrument (Sartorius). All steps

were performed at 25 °C and at an orbital shaking speed of 1000 rpm, and all reagents were formulated in PBSF buffer (1X PBS with 0.1% w/v BSA). All experiments were replicated using the same IgG and Fab preparations. To measure monovalent binding affinities against SARS-CoV-2 RBD variants and SARS-CoV S, recombinant RBDs of SARS-CoV-2 WT (Acro Biosystems, Cat #SPD-C52H3), Beta (Acro Biosystems, Cat #SPD-C52Hp), Delta (Acro Biosystems, Cat #SPD-C52Hh), BA.1 (Acro Biosystems, Cat #SPD-C522f), BA.2 (Acro Biosystems, Cat#SPD-C522g), BA.4/5 (Acro Biosystems, Cat#SPD-C522r), and SARS-CoV (Sino Biological, Cat #40150-V08B2) were biotinylated using EZ-Link™ Sulfo-NHS-LC-Biotin (Thermo Scientific) following manufacturer's recommendations to achieve an average of four biotins per RBD molecule. Biotinylated antigens were diluted (100 nM) in PBSF and loaded onto streptavidin biosensors (Sartorius) to a sensor response of 1.0–1.2 nm and then allowed to equilibrate in PBSF for a minimum of 30 min. After a 60 s baseline step in PBSF, antigen-loaded sensors were exposed (180 s) to 100 nM Fab and then dipped (420 s) into PBSF to measure any dissociation of the antigen from the biosensor surface. Fab binding data with detectable binding responses (>0.1 nm) were aligned, inter-step corrected (to the association step) and fit to a 1:1 binding model using the FortéBio Data Analysis Software (version 11.1).

### ACE2 competition by biolayer interferometry
Antibody binding competition with recombinant human ACE2 receptor (Sino Biological, Cat# 10108-H08H) was determined by BLI using a ForteBio Octet HTX (Sartorius). All binding steps were performed at 25 °C and at an orbital shaking speed of 1000 rpm. All reagents were formulated in PBSF (1X PBS with 0.1% w/v BSA). IgGs (100 nM) were captured onto anti-human IgG capture (AHC) biosensors (Molecular Devices) to a sensor response of 1.0–1.4 nm and then soaked (20 min) in an irrelevant IgG1 solution (0.5 mg/ml) to block remaining Fc binding sites. Next, sensors were equilibrated for 30 min in PBSF and then briefly exposed (90 s) to 300 nM of ACE2 to assess any potential cross interactions between sensor-loaded IgG and ACE2. Sensors were allowed to baseline (60 s) in PBSF before exposing (180 s) to 100 nM SARS-CoV-2 RBD (Acro Biosystems, Cat # SPD-C52H3). Last, RBD-bound sensors were exposed (180 s) to 300 nM ACE2 to assess competition, where antibodies that resulted in increased sensor responses after ACE2 exposure represented non-ACE2-competitive binding profiles, while those resulting in unchanged responses represented ACE2-competitive profiles.

### Deep mutational scanning analysis of antibody binding escape
Yeast-display deep mutational scanning experiments identifying mutations that escape binding by each monoclonal antibody were conducted with duplicate site-saturation mutagenesis Omicron BA.1 RBD libraries[23]. Yeast libraries were grown in SD-CAA media (6.7 g/L Yeast Nitrogen Base, 5.0 g/L Casamino acids, 2.13 g/L MES, and 2% w/v dextrose), and back diluted to 0.67 OD600 in SG-CAA + 0.1%D (SD-CAA with 2% galactose and 0.1% dextrose in place of the 2% dextrose) to induce RBD expression, which proceeded for 16–18 h at room temperature with mild agitation. 5 OD of cells were washed in PBS-BSA (0.2 mg/L) and labeled for 1 h at room temperature in 1 mL with a concentration of antibody determined as the EC90 from pilot isogenic binding assays. In parallel, 0.5 OD of yeast expressing the Omicron BA.1 wildtype RBD were incubated in 100 μL of antibody at the matched EC90 concentration or 0.1x the concentration for FACS gate-setting. Cells were washed and incubated with 1:100 FITC-conjugated chicken anti-Myc antibody (Immunology Consultants CMYC-45F) to label RBD expression and 1:200 PE-conjugated goat anti-human-IgG (Jackson ImmunoResearch, Cat # 109-115-098) to label bound antibody. Labeled cells were washed and resuspended in PBS for FACS.

Antibody-escape cells in each library were selected via FACS on a BD FACSAria II. FACS selection gates were drawn to capture ~50% of yeast expressing the wildtype BA.1 RBD labeled at 10x reduced antibody labeling concentration (see gates in Supplementary Fig. 8d). For each sample, ~4 million RBD⁺ cells were processed on the sorter with a collection of cells in the antibody-escape bin. Sorted cells were grown overnight in SD-CAA + pen-strep, plasmid purified (Zymo D2005), PCR amplified, and barcode sequenced on an Illumina NextSeq. In parallel, plasmid samples were purified from 30 OD of pre-sorted library cultures and sequenced to establish pre-selection barcode frequencies.

Demultiplexed Illumina barcode reads were matched to library barcodes in barcode-mutant lookup tables using dms_variants (version 0.8.9), yielding a table of counts of each barcode in each pre- and post-sort population which is available at https://github.com/jbloomlab/SARS-CoV-2-RBD_Omicron_MAP_Adimab/blob/main/results/counts/variant_counts.csv.gz. The escape fraction of each barcoded variant was computed from sequencing counts in the pre-sort and antibody-escape populations via the formula:

$$E_v = F * \left( \frac{n_v^{\mathrm{post}}}{N^{\mathrm{post}}} \right) \Big/ \left( \frac{n_v^{\mathrm{pre}}}{N^{\mathrm{pre}}} \right)$$

where $F$ is the total fraction of the library that escapes antibody binding, $n_v$ is the counts of variant $v$ in the pre- or post-sort samples with a pseudocount addition of 0.5, and $N$ is the total sequencing count across all variants pre- and post-sort. These escape fractions represent the estimated fraction of cells expressing a particular variant that fall in the escape bin. Per-barcode escape scores are available at https://github.com/jbloomlab/SARS-CoV-2-RBD_Omicron_MAP_Adimab/blob/main/results/escape_scores/scores.csv.

We applied computational filters to remove mutants with low sequencing counts or highly deleterious mutations that had ACE2 binding scores of <−2 or expression scores of <−1, and we removed mutations to the conserved RBD cysteine residues. Per-mutant escape fractions were computed as the average across barcodes within replicates, with the correlations between replicate library selections shown in Supplementary Fig. 8e. Final escape fraction measurements averaged across replicates are available at https://github.com/jbloomlab/SARS-CoV-2-RBD_Omicron_MAP_Adimab/blob/main/results/supp_data/Adimabs_raw_data.csv.

### Statistics and reproducibility
Statistical analyses were performed in GraphPad Prism (version 9.5.1) and R (version 4.2.1). All results are representative of at least two independent replicates. No statistical method was used to pre-determine the sample size. No data were excluded from the analyses. Given the exploratory nature of this study, randomization and blinding were not performed.

### Reporting summary
Further information on research design is available in the Nature Portfolio Reporting Summary linked to this article.

## Data availability
Sequences of antibodies described in this study have been deposited in GenBank (accession codes OQ350107 to OQ350814). Antibody characterization data are available at Zenodo (https://doi.org/10.5281/zenodo.7857455). Omicron BA.1 yeast-display deep mutational scanning libraries are available from Addgene (#1000000187 [https://www.addgene.org/pooled-library/bloom-sars-cov-2-rbd-ssm/]). Spike sequences of SARS-CoV and SARS-CoV-2 were obtained from the ENA (#AAP13441) and GenBank (#NC_045512.2). Raw sequencing data for deep mutational scanning experiments are deposited at the NCBI SRA under BioProject PRJNA770094, BioSample SAMN34380495.

## Materials availability

IgGs are available from L.M.W. under a material transfer agreement from Invivyd Inc. Source data are provided with this paper.

## Code availability

The complete computational pipeline with intermediate and final data files is available from GitHub (https://github.com/jbloomlab/SARS-CoV-2-RBD_Omicron_MAP_Adimab). The code is also deposited at Zenodo (https://doi.org/10.5281/zenodo.7847583).

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

## Acknowledgements

We thank T. Boland for his assistance with sequence analysis. We acknowledge E. Krauland, J. Nett, M. Vasquez, and C. G. Rappazzo for their helpful comments on the manuscript. We thank the Flow Cyto-metry and Genomics Shared Facilities at Fred Hutchinson Cancer Cen-ter. We also thank the DartLab Shared Resources facility at the

Dartmouth Cancer Center. All IgGs were sequenced by Adimab's Molecular Core and produced by the High-Throughput Expression group. Cartoons in Fig. 1a was created with BioRender.com. This study is supported by the NIAID/NIH grants K99AI166250 (T.N.S.) and R01AI141707 (J.D.B.), NCI Cancer Center Support Grant 5P30 CA023108-41 (D.W.M.), Bill and Melinda Gates Foundation INV-004923 (D.R.B.), and the James B. Pendleton Charitable Trust (D.R.B.).

## Author contributions

C.I.K. and L.M.W. conceived and designed the study. D.W.M. supervised and performed clinical sample collection and processing. C.I.K. designed and performed B cell analyses. C.I.K. and P.K. performed single B cell sorting. C.I.K., P.Z., P.K., and H.L.D. performed pseudovirus neutralization assays. T.N.S. designed and performed antibody-deep mutational scanning analyses. C.I.K. and E.R.C. performed biolayer interferometry assays. C.I.K., H.L.D., and G.S. conducted antibody sequence analyses and generated antibody UCAs. C.I.K., T.N.S., H.L.D., J.C.G., D.R.B., R.A., J.D.B., and L.M.W. analyzed the data. C.I.K. and L.M.W. wrote the manuscript, and all authors reviewed and edited the paper.

## Competing interests

C.I.K. and L.M.W. are former employees and hold shares in Adimab. LLC. P.K., H.L.D., E.R.C., and J.C.G. are current employees and hold shares in Adimab LLC. L.M.W. is a former employee and holds shares in Invivyd Inc. T.N.S. and J.D.B. consult with Apriori Bio. J.D.B. has consulted for Moderna and Merck on viral evolution and epidemiology. D.R.B. is a consultant for IAVI, Invivyd, Adimab, Mabloc, VosBio, Nonigenex, and Radiant. C.I.K. and L.M.W. are inventors on a provisional patent application (No. 63/408,980) describing the SARS-CoV-2 antibodies reported in this work. T.N.S. and J.D.B. may receive a share of intellectual property revenue as inventors on Fred Hutchinson Cancer Center–optioned technology and patents related to deep mutational scanning of viral proteins. The remaining authors declare no competing interests.
