## [Peer review file · Nature Communications]

REVIEWER COMMENTS

Reviewer #1 (Remarks to the Author):

Kaku and colleagues investigated the evolution of B cell responses following Omicron breakthrough infections. This is an extension of their previous work, which had reported on responses at 14-27 days post-infection, through 5 months in the current report. The importance of the study is to advance the understanding of how B cell responses evolve over time and whether responses to variants can be elicited de novo. The authors offer answers on both counts, although some of their findings need more clarity.

Specific comments:

1. This is a very small cohort, 4 in one group and 3 in another and for the first group, there is one missing T2. This person should be removed, especially given that paired analyses exclude missing single datapoints.
2. In text beginning line 101 regarding antigen-specific B cells, it is unclear whether the authors compared these for both T1 and T2 in this study or used T1 data from their previous study. If the latter, this would be problematic due to inevitable batching effects. Please clarify and if done at separate times, how did the authors correct for batching?
3. Figure 2E-F: Beginning line 122, the authors state that they isolated 363 antibodies from ~T2 but and compared properties from those isolated at T1. The authors should explicitly state that these were compared to those isolated from their previous study. In addition, while proportional representation from each donor was reasonably similar for the two timepoints and most of the reconstituted antibodies were cross-reactive, what was the contribution from each of the 4 donors to single WT or BA.1 specificities? For BA.1, source of T1 antibodies appears to have been from a single donor (per line 187). Those from T2 appear to be from 4 donors per Fig. S1D. Given the few antibodies reconstituted and skewed sources at T1 and T2 for BA.1, the comparison shown in Figure 2F may not necessarily reflect changes over time. This needs to be addressed.
4. Line 114: the authors state that only 3 of 6 donors had single BA.1-specific B cells, which is supported by Figure S1B and flow plots in S1C. However, the authors then state that single BA.1-specific antibodies were isolated from all 4 donors. Given that the sorting gated on all RBD-binding cells, recovering antibodies with a specificity (BA.1 only) of very low frequency (below detection for donor IML4045) would be expected to occur very rarely if at all. How do the authors explain the apparent disconnect between the specificities measured by flow cytometry and those by antibody-binding assay? A clarification is needed given the importance of understanding the evolution of de novo responses to new variants.

5. From line 188 regarding the expansion of de novo BA.1 responses at T2, the authors state that BA.1 antibodies have SHM levels that are similar to their cross-reactive counterparts. Perhaps, but this statement should be sustained by details on the donor source of BA.1 at T2. Were they a mix from all 4 donors like those with cross-reactivity? Again, this goes to the issue of a skewed representation for BA.1-specific responses.

6. Related to the last point: only 6 of 15 BA.1 antibodies neutralized BA.1 in Figure 4D. How does this relate to their SHM and similarities in SHM with the other antibodies, all of which neutralized BA.1 < 1.0 per Figure 3E? Was there anything unique in gene usage of these BA.1 antibodies?

Minor points:

1. Figure 1: the authors indicate IC50 for serum neutralization but it should be written as ID50 if based on dilution. In Figure 1D, the authors show fold change in neutralization and in text write fold reduction of variant over D614G. While perhaps technically correct, the graph is likely to be interpreted as variant having a higher ID50 than D614G, when in fact it is the reverse. Perhaps the graph should indicate D614G/variant and modify text accordingly.

2. Figure 2D: clarify that the p value shown refer to changes for WT specificity if indeed that is what the color coding means.

Reviewer #2 (Remarks to the Author):

Kaku and colleagues investigate the humoral response elicited by Omicron BA.1 infection in individuals who received two or three vaccine doses. The authors analyzed serum neutralization and B cell memory. Beyond the usual neutralization assays, the methods are of high standards, with BLI measurements, antibodies cloning and deep-mutational scanning. The authors confirmed induction of cross-reactive and omicron-specific B cell clones upon BA.1 breakthrough infection. Interestingly, they suggest a mechanism to explain the convergent evolution of current Omicron sub-variants such as BA.2.75.2, BA.4.6, BQ1.1... They propose that the preferential recall of public B cell clones that are cross-reactive between BA.1 and ancestral viruses have narrowed the humoral response to some conserved Wuhan/BA.1 epitopes, which in turn became preferential mutational hotspots. This model is interesting, supported by the data and potentially important to guide vaccine design and predict future emergence of variants. I only have minor comments.

Specific concerns:

1. For clarity's sake, I would suggest reorganizing figure 1, to sort variants according to their extent of immune evasion.
2. What are the clonal relationships between memory B cells characterized at T1 and T2?
3. Is it possible to add pre-BT samples to confirm that cross-reactive clones were indeed pre-existing?
4. Is there any notable difference between individuals who received 2 or 3 vaccine doses?
5. BQ.1.1 should be added to the list of emerging variants in the introduction (line 46)
6. Line 58, the authors should mitigate the observation that variant-containing vaccines are only modestly more potent than monovalent vaccines. Additional reports suggest a limited immune imprinting and an advantage conferred by bivalent vaccines.
7. What are the epitopes of de novo BA.1-specific antibodies?
8. In the absence of a control cohort (BA.1 infection in unvaccinated individuals) it is impossible to know whether the preferential induction of cross-reactive BA.1/WT B cell clone is due to imprinting immunity. This should be discussed.

Reviewer #3 (Remarks to the Author):

In the manuscript Kaku and colleagues describe the antibody response changes after vaccination and consecutive breakthrough infection in 6(7) Individuals and show a continued evolution after exposure to Omicron BA1. The manuscript is concise, well-written and experiments are well performed. However, there are a couple of major issues that I would like to see addressed:

1.) The study is based on 6 breakthrough infections where individuals received either two or three vaccination and then had a breakthrough infection. This reduces the group to 3 each respectively. In addition time since blood draws are between 14 and 27 or 122 and 170 days. Given the very low number of participants, the differences in vaccination schemes and blood draw, I would caution to draw meaningful and strong conclusions. Ideally the authors would increase the sample size significantly.

2.) As the authors describe a narrow breadth of the BA-1-specific de novo response as well improved activity towards the RBD domain, it would be essential to show a comparison to omicron only (!) infected individuals. In particular as the author argue that BA1 breakthrough infection favors further evolution of pre-existing B cell memory over de novo activity, but do not show any data to back this up.

3.) One would think of an experiment where the vaccine induced responses are depleted and only Omicron specific responses are being compared.

3.) The authors describe a waning of the B cell immune responses. This is not unusual. Did the authors address whether this is particularly rapid in comparison to unvaccinated individuals?

Minor:

For transparency reasons it would be important to understand the role of both pharmaceutical companies in this manuscript.

Dear Dr. Pickford,

We were pleased to receive such positive feedback from all reviewers, and we found their comments and suggestions to be helpful in improving our manuscript. We have addressed each specific concern and have modified the manuscript accordingly.

Reviewer #1:

1. This is a very small cohort, 4 in one group and 3 in another and for the first group, there is one missing T2. This person should be removed, especially given that paired analyses exclude missing single datapoints.

We thank the reviewer for this suggestion and have now removed this individual from analysis.

2. In text beginning line 101 regarding antigen-specific B cells, it is unclear whether the authors compared these for both T1 and T2 in this study or used T1 data from their previous study. If the latter, this would be problematic due to inevitable batching effects. Please clarify and if done at separate times, how did the authors correct for batching?

We thank the reviewer for raising this point. We used B cell staining data collected from our previous study. To maintain consistency in our FACS analysis between the early and late time points, we labeled B cells with recombinant antigen aliquots originating from the same protein preparation. We further corrected for any non-specific background binding at both time points by including the same negative control sample collected prior to the COVID-19 pandemic. We have now clarified this detail in the figure legends.

3. Figure 2E-F: Beginning line 122, the authors state that they isolated 363 antibodies from ~T2 but compared properties from those isolated at T1. The authors should explicitly state that these were compared to those isolated from their previous study. In addition, while proportional representation from each donor was reasonably similar for the two timepoints and most of the reconstituted antibodies were cross-reactive, what was the contribution from each of the 4 donors to single WT or BA.1 specificities? For BA.1, source of T1 antibodies appears to have been from a single donor (per line 187). Those from T2 appear to be from 4 donors per Fig. S1D. Given the few antibodies reconstituted and skewed sources at T1 and T2 for BA.1, the comparison shown in Figure 2F may not necessarily reflect changes over time. This needs to be addressed.

We thank the reviewer for these suggestions. We have modified the text to clarify that antibodies isolated from the late time point are compared to those reported in the previous study, as shown below:

Similar to the previously characterized antibodies from the acute time point, most of the newly isolated antibodies recognized both WT and BA.1 RBD antigens (73-97%), and we observed a bias toward certain VH germline genes.

We isolated WT-specific antibodies from all four donors at both time points, ranging from 1-6 and 2-11 antibodies per individual from the acute and longitudinal samples,

respectively. While we identified BA.1-specific antibodies from only one donor at the early time point, we isolated 2-8 BA.1-specific antibodies from each of the four donors at the late time point. We acknowledge the caveat that the small number of BA.1-specific antibodies isolated from the 1-month time point is a limited representation of the properties of the early BA.1-specific response. However, the lack of WT RBD recognition by these antibodies suggest that they were induced as a de novo, primary response to Omicron infection, which is consistent with the low levels of somatic hypermutations (SHM) observed at two weeks post convalescence. Given that BA.1-specific antibodies are virtually absent from the acute B cell response to infection, we focus our analysis on the sequence and functional activities of BA.1-specific antibodies derived from the late time point. Here, antibodies isolated from all four donors exhibited significant levels of SHM (median = 11 nucleotides), suggesting that BA.1-specific antibodies underwent cycles of SHM in germinal center reactions. We have now removed the direct comparison of SHM over time, as shown below:

Additionally, the level of SHM in the cross-reactive antibodies increased from a median of 9 VH nucleotide substitutions at 1-month to 11 VH nucleotide substitutions by 5-6 months, potentially suggesting that BA.1 breakthrough infection drives further affinity maturation of pre-existing cross-reactive memory B cells (Fig. 2f).

4. Line 114: the authors state that only 3 of 6 donors had single BA.1-specific B cells, which is supported by Figure S1B and flow plots in S1C. However, the authors then state that single BA.1-specific antibodies were isolated from all 4 donors. Given that the sorting gated on all RBD-binding cells, recovering antibodies with a specificity (BA.1 only) of very low frequency (below detection for donor IML4045) would be expected to occur very rarely if at all. How do the authors explain the apparent disconnect between the specificities measured by flow cytometry and those by antibody-binding assay? A clarification is needed given the importance of understanding the evolution of de novo responses to new variants.

We thank the reviewer for raising this point. While we assessed the binding specificity of monoclonal antibodies via biolayer interferometry (BLI) using recombinant RBD antigens, FACS-based analysis was performed by staining B cells with RBD tetramers, which represents a multi-avid environment due to the multivalent nature of tetramerized antigen preparation and the potential for multiple B cell receptors to cross-link antigen tetramers on the cell surface. Therefore, it is possible that antibodies may bind WT RBD tetramers under highly avid conditions even if this is undetectable under the conditions used for BLI.

We agree with the reviewer that characterization of BA.1-specific B cells is critical to our understanding of the magnitude and specificities of de novo responses induced by heterologous variant exposure. To further confirm that BA.1-specific B cells were newly primed by infection, we randomly selected four clonally-distinct BA.1-specific antibodies and produced their inferred unmutated common ancestors (UCAs) as recombinant IgGs. All four UCAs showed reduced binding to WT RBD antigens relative to BA.1, consistent with the notion that BA.1-specific B cells represent a de novo response to infection. We have now included this data in Supplementary Fig. 4 and commented on this in the text.

5. From line 188 regarding the expansion of de novo BA.1 responses at T2, the authors state that BA.1 antibodies have SHM levels that are similar to their cross-reactive counterparts. Perhaps, but this statement should be sustained by details on the donor source of BA.1 at T2. Were they a mix from all 4 donors like those with cross-reactivity? Again, this goes to the issue of a skewed representation for BA.1-specific responses.

Thank you for this suggestion. As noted in our response to comment #3, our analysis includes BA.1-specific antibodies from all four donors at the 5-6-month time point. Thus, we believe that analysis of the pooled set of BA.1-specific antibodies is fairly representative of BA.1-specific memory at 5 to 6 months following breakthrough infection. We have now updated the text with this detail and have also toned down our comparison of SHM with those of cross-reactive antibodies, as shown in our comment #3 response.

6. Related to the last point: only 6 of 15 BA.1 antibodies neutralized BA.1 in Figure 4D. How does this relate to their SHM and similarities in SHM with the other antibodies, all of which neutralized BA.1 < 1.0 per Figure 3E? Was there anything unique in gene usage of these BA.1 antibodies?

We thank the reviewer for these questions. We did not observe significant differences in SHM levels between neutralizing and non-neutralizing antibodies for either BA.1-specific or WT/BA.1 cross-reactive antibodies (Figure 1 for reviewer). We also did not observe notable germline gene usage patterns among BA.1-specific antibodies, although we are hesitant to draw strong conclusions given the small number of BA.1-specific antibodies isolated.

Figure 1 for reviewer:

Minor points:

1. Figure 1: the authors indicate IC50 for serum neutralization but it should be written as ID50 if based on dilution. In Figure 1D, the authors show fold change in neutralization and in text write fold reduction of variant over D614G. While perhaps technically correct, the graph is likely to be interpreted as variant having a higher ID50 than D614G, when in fact it is the reverse. Perhaps the graph should indicate D614G/variant and modify text accordingly.

We apologize for the confusion and have now corrected serum neutralizing titers to ID50 and Fig. 1d to “D614G/variant.”

2. Figure 2D: clarify that the p value shown refer to changes for WT specificity if indeed that is what the color coding means.

We thank the reviewer for this suggestion and have clarified this point in the legends.

Reviewer #2:

1. For clarity’s sake, I would suggest reorganizing figure 1, to sort variants according to their extent of immune evasion.

We thank the reviewer for this suggestion. Since the study donors were exposed to the ancestral strain (through vaccination) and BA.1 (through breakthrough infection), we believe that comparisons between the two strains should be prioritized in our analysis and thus placed in proximity to each other. Further, pre-Omicron variants such as Beta and Delta are no longer in circulation, while new Omicron sublineages have now emerged and spread across the population. Given the increased relevance of Omicron sublineages compared with pre-Omicron variants as well as their antigenic similarity to BA.1, we believe it is appropriate to order the variants as shown in Fig. 1.

2. What are the clonal relationships between memory B cells characterized at T1 and T2?

We thank the reviewer for this question. The proportion of persistent clonal lineages ranged from 4-30% of the RBD-directed antibody repertoire at the late time point (Figure 2 for reviewer). We note that antibodies isolated 5-6-month post-infection do not necessarily imply delayed emergence in the peripheral memory compartment (eg. a clone identified at a late time point may have developed at an earlier point). Thus, we are hesitant to draw strong conclusions from sequence and binding analyses of persistent clones in the absence of single-cell fate mapping experiments. However, we have now included this data as Supplementary Fig. 3 and commented on it in the text.

Figure 2 for reviewer:

3. Is it possible to add pre-BT samples to confirm that cross-reactive clones were indeed pre-existing?

We thank the reviewer for this suggestion. Unfortunately, we were unable to collect blood samples prior to breakthrough infection given the unpredictable nature of infection. However, cross-reactive antibodies isolated from the 1-month time point displayed levels of somatic hypermutation comparable to those from 6 months post-infection, suggesting that these antibodies originate from the memory B cell compartment rather than naïve B cells primed by breakthrough infection¹. Consistent with this notion, the vast majority (>95%) of cross-reactive antibodies at this time point displayed biased binding to WT relative to BA.1 RBD (Fig. 3a).

While cross-reactive antibodies isolated at the late time point display higher median levels of SHM compared with those from the 1-month time point (Fig. 2f), we cannot rule out the possibility that newly primed B cells with WT/BA.1 cross-reactivity may have accumulated SHMs over the course of 5-6-months without performing deep BCR sequencing of pre- and post-infection samples. However, we note that approximately 50% of antibodies still display skewed binding to the WT RBD, suggesting a vaccine-mediated imprinting response (Fig. 3c). For ten BA.1-preferring antibodies, we have now produced unmutated common ancestors (UCAs) for a selection of such antibodies to assess whether their UCAs displayed stronger WT RBD binding. To maximize the diversity of our panel, we included antibodies from all four donors and excluded any that were clonally related. 9 of the 10 UCAs displayed preferential binding to the WT RBD, providing further evidence of a vaccine-induced origin. We have now included this data in Supplementary Fig. 4 and discussed these results in the text.

4. Is there any notable difference between individuals who received 2 or 3 vaccine doses?

Individuals who received three-doses of the vaccine generally display higher serum neutralization titers and higher frequencies of RBD-specific cells among class-switched B cells compared with those who experienced breakthrough infection after two-doses. We have indicated two-dose and three-dose vaccine recipients in Figure 1 and 2 via circles and triangles, respectively. This is consistent with the overall larger magnitude of humoral responses observed after a third vaccine shot irrespective of subsequent infection^{2,3}.

5. BQ.1.1 should be added to the list of emerging variants in the introduction (line 46)

Thank you for this suggestion. We have now added BQ,1.1 as well as XBB to the list of emerging variants.

6. Line 58, the authors should mitigate the observation that variant-containing vaccines are only modestly more potent than monovalent vaccines. Additional reports suggest a limited immune imprinting and an advantage conferred by bivalent vaccines.

We thank the reviewer for this suggestion. Assessment of the bivalent mRNA-1273.214 (encoding Wuhan-1 and BA.1 S proteins) revealed a 1.3- to 2-fold higher serum neutralizing titers against BA.1 one month after bivalent booster immunization compared with the ancestral mRNA-1273 vaccine⁴. Consistent with this result, a separate study

showed that bivalent mRNA-1273.222 (encoding Wuhan-1 and BA.5 S proteins) elicited 2.3-fold higher titers against BA.5 and 1.5-2.5-fold higher titers against BA.2.75.2, BQ.1.1, and XBB compared with the Wuhan-1 monovalent vaccine⁵. Finally, comparison of individuals receiving fourth monovalent dose with those receiving three monovalent doses and a fourth bivalent booster by David Ho's group suggested no significant differences in serum neutralizing titers across a panel of SARS-CoV-2 variants⁶. Considering the totality of published evidence, we believe that the benefit of bivalent vaccine appears relatively modest relative to the monovalent vaccine, at least for serum responses within one month of vaccination. The latter two studies were not published at the time of manuscript submission, and we have now included these references in the revised text.

To our knowledge, the only analysis of B cell responses following variant-based booster vaccination is a recent preprint from Ali Ellebeddy's group studying donors who received mRNA-1273.529, a monovalent vaccine encoding only the BA.1 S protein⁷. The authors demonstrate that 99% of isolated antibodies displayed cross-reactivity with both BA.1 and WT S, thus also suggesting a strong degree of imprinting by the ancestral vaccine. We have now commented on this result in the text.

7. What are the epitopes of de novo BA.1-specific antibodies?

We thank the reviewer for this question. BA.1-specific antibodies could only be detected in 1 of 5 donors at the 1-month time point and despite a modest increase in frequency by 5-6-months post-infection, comprise on average only 4% of the overall BA.1-reactive B cell response (1-15% per donor) (Supplementary Fig. 1d). Given that BA.1-specific antibodies comprise only a small fraction of the memory compartment, we have not performed a formal analysis of their epitopes via deep mutational scanning or structural characterization. However, analysis of their breadth of binding against a panel of 5 SARS-CoV-2 RBD variants and the SARS-CoV RBD revealed that 3/6 BA.1-specific neutralizing antibodies failed to bind BA.2 RBD, suggesting that residues uniquely mutated in BA.2 (eg. D405N, R048S) may have contributed to their binding epitope (Fig. 4d). Similarly, the remaining three BA.2/BA.1-cross-reactive antibodies failed to recognize the BA.5 RBD, suggesting that amino acid positions mutated in BA.5 (eg. L452R, F486V) may play a role in antibody recognition (Fig. 4d).

8. In the absence of a control cohort (BA.1 infection in unvaccinated individuals) it is impossible to know whether the preferential induction of cross-reactive BA.1/WT B cell clone is due to imprinting immunity. This should be discussed.

We thank the reviewer for pointing this out. We acknowledge that we cannot definitively conclude whether cross-reactive B cells isolated from the 5-6-month time point were primed by vaccination or later by breakthrough infection. However, our UCA analysis demonstrating the biased binding of inferred germline antibodies to WT RBD relative BA.1 RBD suggests that cross-reactive antibodies most likely originate from pre-existing memory rather than naïve B cells primed by breakthrough infection. Consistent with this notion, studies of B cell responses following primary Omicron infection have shown that

the majority of RBD-directed B cells fail to recognize the ancestral RBD, suggesting that the majority of cross-reactive B cells are unlikely to have been primed by Omicron exposure⁸⁻¹⁰. However, we have now softened the conclusions related to the origin of these cross-reactive antibodies.

Reviewer #3:

1.) The study is based on 6 breakthrough infections where individuals received either two or three vaccination and then had a breakthrough infection. This reduces the group to 3 each respectively. In addition time since blood draws are between 14 and 27 or 122 and 170 days. Given the very low number of participants, the differences in vaccination schemes and blood draw, I would caution to draw meaningful and strong conclusions. Ideally the authors would increase the sample size significantly.

We thank the reviewer for this suggestion. Although we agree that this study would benefit from a higher sample size, unfortunately, we were unable to find and consent additional donors with BA.1 breakthrough infection for inclusion in the study. However, our observations that the B cell response following BA.1 breakthrough infection is primarily mediated by the recall of vaccine-induced memory and the limited magnitude of the de novo response appears consistent with studies of the acute B cell response to Omicron exposure through either infection or mRNA-1273.529 immunization^{1,7,11,12}. We have also acknowledged the small sample size in the text, as shown below:

At the late time point, we also detected the emergence of a BA.1-specific B cell population in 3 of the 6 individuals (averaging 3% of class-switched RBD⁺ B cells), although this increase in BA.1-specific cells did not reach statistical significance due to the small cohort size and variability in the magnitude of this response among individuals (ranging from 1-18%) (Fig. 2d, Supplementary Fig. 1b).

2.) As the authors describe a narrow breadth of the BA.1-specific de novo response as well improved activity towards the RBD domain, it would be essential to show a comparison to omicron only (!) infected individuals. In particular as the author argue that BA1 breakthrough infection favors further evolution of pre-existing B well memory over de novo activity, but do not show any data to back this up.

We thank the reviewer for pointing this out. While we agree that a comparison of BA.1-specific antibodies elicited by breakthrough versus primary infection could be interesting, we were unable to find Omicron infected donors with no prior infection or vaccination. However, we have shown that BA.1-specific antibodies, which would not be elicited in the absence of BA.1 infection, comprise only a small proportion of the total RBD-directed response. Further, studies have shown that primary Omicron infection elicits serum and B cell responses with limited cross-reactivity with the WT strain, suggesting that BA.1-specific antibodies likely represent the majority of the de novo response⁸⁻¹⁰. Although we acknowledge our lack of direct evidence of the origins of WT/BA.1 cross-reactive antibodies, we reasoned that if cross-reactive antibodies, including those with preferential binding to BA.1 compared with WT, were induced by vaccination, their inferred germline sequences should show higher reactivity to the WT RBD. We produced UCAs for ten BA.1-preferring antibodies and characterized their

binding affinity to WT and BA.1 RBDs. Consistent with our hypothesis of a recall origin, 9/10 UCAs displayed higher affinity binding to the WT RBD. We have now included this data in Supplementary Fig. 4 and discussed these results in the text. We have also toned down our conclusions and included these caveats in the discussion section.

3.) One would think of an experiment where the vaccine induced responses are depleted and only Omicron specific responses are being compared.

We thank the reviewer for this suggestion. Serum depletion using WT RBD will remove all WT-reactive antibodies and isolate BA.1-specific antibodies, but we believe this experiment does not offer insight into whether the antibodies originate from B cells primed by the vaccine versus by infection. While primary Omicron infection in unvaccinated individuals have been shown to elicit 15-fold higher serum neutralizing titers against Omicron compared with the ancestral strain, donors exhibited, on average neutralizing titers of 1:96 against the WA-1 strain⁹. Therefore, depletion with a WT antigen will remove all vaccine-induced antibodies as well as a small fraction of de novo induced antibodies. Direct comparison of vaccine- and infection-elicited serum responses would require Ig-Seq analysis paired with deep B cell sequencing of samples before and after breakthrough infection, which is beyond the scope of our study.

4.) The authors describe a waning of the B cell immune responses. This is not unusual. Did the authors address whether this is particularly rapid in comparison to unvaccinated individuals?

We thank the reviewer for raising this point. Infection in unvaccinated individuals induces a primary humoral response to SARS-CoV-2, where frequencies of antigen-specific B cell increase for up to 6 months post-infection as naïve B cells mature and develop into memory B cells (MBCs)¹³⁻¹⁵. In contrast, subsequent exposures reactivates pre-existing MBCs, which rapidly proliferate and differentiate into plasmablasts or re-enter the germinal center to undergo further rounds of affinity maturation. While B cell responses increase over time in previously unexposed individuals, breakthrough infection represents a secondary response, which is characterized by the differentiation of plasmablasts from vaccine-induced MBCs, which then wane over time following the contraction of this response. Similarly, antigen-specific B cells have been shown to decline 1.5-2.5-fold over the course of 3 months after a third mRNA vaccine dose, although we are unable to directly compare the rate of decline due to different lengths of study periods (3-month versus 5-6-months)¹⁶.

Minor:

For transparency reasons it would be important to understand the role of both pharmaceutical companies in this manuscript.

The contributions and competing interested of all the authors are listed at the end of the references section.

We hope that the modifications to the manuscript meet your approval.

Best wishes,
Laura M. Walker

References:

1. Kaku, C. I. *et al.* Recall of preexisting cross-reactive B cell memory after Omicron BA.1 breakthrough infection. *Sci. Immunol.* **7**, eabq3511 (2022).
2. Muecksch, F. *et al.* Increased memory B cell potency and breadth after a SARS-CoV-2 mRNA boost. *Nature* **607**, 128–134 (2022).
3. Goel, R. R. *et al.* Efficient recall of Omicron-reactive B cell memory after a third dose of SARS-CoV-2 mRNA vaccine. *Cell* **185**, 1875-1887.e8 (2022).
4. Chalkias, S. *et al.* A Bivalent Omicron-Containing Booster Vaccine against Covid-19. *N. Engl. J. Med.* **387**, 1279–1291 (2022).
5. Davis-Gardner, M. E. *et al.* Neutralization against BA.2.75.2, BQ.1.1, and XBB from mRNA Bivalent Booster. *N. Engl. J. Med.* (2022) doi:10.1056/NEJMc2214293.
6. Wang, Q. *et al.* Antibody response to omicron BA.4–BA.5 bivalent booster. *N. Engl. J. Med.* (2023) doi:10.1056/NEJMc2213907.
7. Alsoussi, W. B. *et al.* SARS-CoV-2 Omicron boosting induces de novo B cell response in humans. *BioRxiv* (2022) doi:10.1101/2022.09.22.509040.
8. Richardson, S. I. *et al.* SARS-CoV-2 Omicron triggers cross-reactive neutralization and Fc effector functions in previously vaccinated, but not unvaccinated, individuals. *Cell Host Microbe* **30**, 880-886.e4 (2022).
9. Suryawanshi, R. K. *et al.* Limited cross-variant immunity from SARS-CoV-2 Omicron without vaccination. *Nature* **607**, 351–355 (2022).
10. Cao, Y. *et al.* BA.2.12.1, BA.4 and BA.5 escape antibodies elicited by Omicron infection. *Nature* **608**, 593–602 (2022).
11. Quandt, J. *et al.* Omicron BA.1 breakthrough infection drives cross-variant neutralization and memory B cell formation against conserved epitopes. *Sci. Immunol.* **7**, eabq2427 (2022).
12. Wang, Z. *et al.* Memory B cell responses to Omicron subvariants after SARS-CoV-2 mRNA breakthrough infection in humans. *J. Exp. Med.* **219**, (2022).
13. Sakharkar, M. *et al.* Prolonged evolution of the human B cell response to SARS-CoV-2 infection. *Sci. Immunol.* **6**, (2021).
14. Wang, Z. *et al.* Naturally enhanced neutralizing breadth against SARS-CoV-2 one year after infection. *Nature* **595**, 426–431 (2021).
15. Sokal, A. *et al.* Maturation and persistence of the anti-SARS-CoV-2 memory B cell response. *Cell* **184**, 1201-1213.e14 (2021).
16. Goel, R. R. *et al.* mRNA vaccines induce durable immune memory to SARS-CoV-2 and variants of concern. *Science* **374**, abm0829 (2021).

REVIEWER COMMENTS

Reviewer #1 (Remarks to the Author):

Overall, the authors have provided satisfactory answers to reviewer questions and revised their manuscript accordingly. However, the following points should be addressed.

Response to #4

The authors produced 4 UCAs from BA.1 to evaluate de novo responses, finding reduced binding to WT compared to BA.1 RBD and concluding that this is consistent with a de novo response to infection. In response to Point #3 of Reviewer 2, the authors show the reverse for UACs of cross-reactive antibodies. The authors refer to a new Supplementary Fig. 4 in both responses and only seem to refer to the latter UACs in the Results. Where in the text do they comment on BA.1-specific B cells likely being from a de novo response based on the 4 UCAs generated from antibodies that only bound BA.1? It would be helpful to reviewers if authors would state or track where they make changes to text.

Response to #5

Why not include the reviewer figure in the paper as it is likely to be of interest to readers, even if conclusions are limited? However, the reviewer figure raises a new question that should be addressed: The authors show a high number of non-neuts among cross-reactive antibodies in the reviewer figure yet this does not seem to be the case in Fig. 3F. How do the authors define non-neutralizing antibodies? Only a handful are above 1 $\mu\text{g}/\text{ml}$ in Figure 3F for cross-reactive antibodies, far fewer than shown in the reviewer figure. This should be clarified and included in the revised manuscript with interpretation.

Minor point:

Supplemental Figure 1a: the authors have BA.1-specific written twice of the lower left plot. Should the one in the lower right quadrant be WT-specific as shown if same plot of Fig 2a?

Reviewer #2 (Remarks to the Author):

The authors have addressed my concerns

Reviewer #3 (Remarks to the Author):

Unfortunately, the authors did not respond to the reviews at all. Findings in the manuscript are based on either small numbers or previous published literature. Some findings are anecdotal at best.

Dear Dr. Pickford,

We found the reviewer comments and suggestions to be helpful in improving our manuscript and have now addressed each specific concern and modified the manuscript accordingly.

Reviewer follow-up comments:

Reviewer #1:

1. The authors produced 4 UCAs from BA.1 to evaluate de novo responses, finding reduced binding to WT compared to BA.1 RBD and concluding that this is consistent with a de novo response to infection. In response to Point #3 of Reviewer 2, the authors show the reverse for UACs of cross-reactive antibodies. The authors refer to a new Supplementary Fig. 4 in both responses and only seem to refer to the latter UACs in the Results. Where in the text do they comment on BA.1-specific B cells likely being from a de novo response based on the 4 UCAs generated from antibodies that only bound BA.1? It would be helpful to reviewers if authors would state or track where they make changes to text.

We apologize for the confusion and thank the reviewer for pointing this out. We have now added text referencing the UCA results for both the BA.1-specific (lines 221-228) and cross-reactive antibodies (lines 157-162 in the text).

2. Why not include the reviewer figure in the paper as it is likely to be of interest to readers, even if conclusions are limited? However, the reviewer figure raises a new question that should be addressed: The authors show a high number of non-neuts among cross-reactive antibodies in the reviewer figure yet this does not seem to be the case in Fig. 3F. How do the authors define non-neutralizing antibodies? Only a handful are above 1 $\mu\text{g}/\text{ml}$ in Figure 3F for cross-reactive antibodies, far fewer than shown in the reviewer figure. This should be clarified and included in the revised manuscript with interpretation.

We thank the reviewer for these suggestions. We have now included this data as Supplementary Fig. 5 and commented on the results for the WT/BA.1 cross-reactive antibodies in the text (line 168-170). Due to the low number of BA.1-specific antibodies, we were unable to determine whether or not there are significant differences in the levels of somatic hypermutation in neutralizing versus non-neutralizing BA.1-specific antibodies, but we have still included this data in Supplementary Fig. 5 for reference.

*The high number of non-neutralizing antibodies in the reviewer figure compared to Fig. 3F is because all **cross-binding** antibodies are shown in the reviewer figure (including both neutralizing and non-neutralizing), whereas only **D614G/BA.1 cross-neutralizing** antibodies are shown in Fig. 3D-F. Antibodies that did not reach IC_{50} at a concentration of 2 $\mu\text{g}/\text{ml}$ are defined as non-neutralizing. We have now included this cut-off in the text and in the legends for Fig. 3D-F, the related Supplementary Fig. 6, and Supplementary Fig. 5 (previously the reviewer figure).*

3. Supplemental Figure 1a: the authors have BA.1-specific written twice of the lower left plot. Should the one in the lower right quadrant be WT-specific as shown if same plot of Fig 2a?

We thank the reviewer for pointing this out. We have now corrected the label in Supplementary Fig. 1A to “WT-specific.”

Reviewer #3:

1. Unfortunately, the authors did not respond to the reviews at all. Findings in the manuscript are based on either small numbers or previous published literature. Some findings are anecdotal at best.

We thank the reviewer for their feedback. We agree with the reviewer that ideally we would have included a larger number donors in this study. However, given limitations in the number of antibodies we could practically clone, express, and characterize, we chose to isolate a large number of antibodies from a small number donors rather than a limited number of antibodies from a large number of donors. As such, we were able to draw more definitive conclusions about the evolution of the B cell response within donors than across the donor cohort. Thus, as stated by the reviewer, caution should be exercised in the generalization of these conclusions, which we have now commented on in the discussion.

We acknowledge that we have included previously published data from the same cohort for the purposes of comparing the antibody response at early (prior study) and late (this study) time points following BA.1 breakthrough infection. The results of this study would be difficult for readers to interpret in the absence of the side-by-side comparison with the previously published early timepoint data, so we would prefer not to exclude these data from the manuscript. We have referenced the prior study data in the text, and we have now also included this citation where appropriate in the figure legends.

We have now included a section in the discussion section addressing these limitations as well as others raised by the reviewer originally. We have also discussed these revisions point-by-point below.

1.) The study is based on 6 breakthrough infections where individuals received either two or three vaccination and then had a breakthrough infection. This reduces the group to 3 each respectively. In addition time since blood draws are between 14 and 27 or 122 and 170 days. Given the very low number of participants, the differences in vaccination schemes and blood draw, I would caution to draw meaningful and strong conclusions. Ideally the authors would increase the sample size significantly.

We thank the reviewer for this suggestion. We agree that this study would benefit from a larger sample size and reduced variability in the timing of blood draws and vaccination history of the donors prior to breakthrough infection. We have now included a limitations

section in the discussion explicitly addressing the small sample size, demographics biases of our cohort, as well as variability in the vaccination histories and timing of blood sample collections.

2.) As the authors describe a narrow breadth of the BA-1-specific de novo response as well improved activity towards the RBD domain, it would be essential to show a comparison to omicron only (!) infected individuals. In particular as the author argue that BA1 breakthrough infection favors further evolution of pre-existing B well memory over de novo activity, but do not show any data to back this up.

We thank the reviewer for raising this concern. While we agree that a comparison of BA.1-specific antibodies elicited following breakthrough versus primary infection would be interesting, we were unable to identify Omicron infected donors with no prior infection or vaccination due to the high rates of both vaccination and infection across the population. To address questions regarding the origin of WT/Omicron BA.1 cross-reactive antibodies, we produced unmutated common ancestors (UCAs) for ten BA.1-preferring antibodies and characterized their binding affinity to WT and BA.1 RBDs. If these cross-reactive antibodies were originally induced by prior vaccination, their inferred germline precursors should display comparable or stronger binding to the WT RBD. Indeed, nine of the 10 UCA antibodies displayed higher affinity binding to the WT RBD relative to the BA.1 RBD, suggesting a vaccine-induced memory B cell origin. We have now included this data in Supplementary Fig. 4 and discussed these results in the text. We have also toned down our conclusions regarding the origin of the cross-reactive clones (as we acknowledge limitations in the UCA analysis) and included the lack of availability of samples from unvaccinated, primary BA.1-infected donors in the limitations section of the discussion.

3.) One would think of an experiment where the vaccine induced responses are depleted and only Omicron specific responses are being compared.

We thank the reviewer for this suggestion. Unfortunately, due to the limited amounts of remaining sera from these individuals we are unable to perform this depletion experiment. However, we have now included this as a limitation of our study in the discussion section.

4.) The authors describe a waning of the B cell immune responses. This is not unusual. Did the authors address whether this is particularly rapid in comparison to unvaccinated individuals?

We thank the reviewer for raising this point. In contrast to the results observed in our current study, previous studies (including one from own group) have shown that frequencies of antigen-specific memory B cells increase for up to 6 months following primary SARS-CoV-2 infection¹⁻³. The reasons for this discrepancy are unclear but may be due to the increased magnitude of the initial short-lived B cell response and/or reduced germinal center size or longevity following secondary viral exposure. We have now commented on this in the results section.

We hope that the modifications to the manuscript meet your approval.

Best wishes,
Laura M. Walker

References:

1. Sakharkar, M. *et al.* Prolonged evolution of the human B cell response to SARS-CoV-2 infection. *Sci. Immunol.* **6**, (2021).
2. Wang, Z. *et al.* Naturally enhanced neutralizing breadth against SARS-CoV-2 one year after infection. *Nature* **595**, 426–431 (2021).
3. Sokal, A. *et al.* Maturation and persistence of the anti-SARS-CoV-2 memory B cell response. *Cell* **184**, 1201-1213.e14 (2021).

REVIEWERS' COMMENTS

Reviewer #1 (Remarks to the Author):

The authors have appropriately addressed the last set of concerns.